# A CPC-shelterin-BTR axis regulates mitotic telomere deprotection

Diana Romero-Zamora [1,2,5], Samuel Rogers[3,5], Ronnie Ren Jie Low [3], Scott G. Page [3], Blake J. E. Lane[3], Shunya Kosaka[1,2], Andrew B. Robinson[3], Lucy French [3], Noa Lamm[3], Fuyuki Ishikawa[1], Makoto T. Hayashi[1,2,4] ✉ & Anthony J. Cesare [3] ✉

Telomeres prevent ATM activation by sequestering chromosome termini within telomere loops (t-loops). Mitotic arrest promotes telomere linearity and a localized ATM-dependent telomere DNA damage response (DDR) through an unknown mechanism. Using unbiased interactomics, biochemical screening, molecular biology, and super-resolution imaging, we found that mitotic arrest-dependent (MAD) telomere deprotection requires the combined activities of the Chromosome passenger complex (CPC) on shelterin, and the BLM-TOP3A-RMI1/2 (BTR) complex on t-loops. During mitotic arrest, the CPC component Aurora Kinase B (AURKB) phosphorylated both the TRF1 hinge and TRF2 basic domains. Phosphorylation of the TRF1 hinge domain enhances CPC and TRF1 interaction through the CPC Survivin subunit. Meanwhile, phosphorylation of the TRF2 basic domain promotes telomere linearity, activates a telomere DDR dependent on BTR-mediated double Holliday junction dissolution, and leads to mitotic death. We identify that the TRF2 basic domain functions in mitosis-specific telomere protection and reveal a regulatory role for TRF1 in controlling a physiological ATM-dependent telomere DDR. The data demonstrate that MAD telomere deprotection is a sophisticated active mechanism that exposes telomere ends to signal mitotic stress.

Telomeres are the specialized nucleoprotein structures at eukaryotic chromosome termini. The paramount telomere activity is protection of chromosome ends through localized regulation of DDR and DNA repair activity by the telomere-specific shelterin protein complex[1]. Telomere deprotection refers to physiological outcomes where chromosome end protection is compromised through biological processes, typically in the presence of wild type shelterin[2].

Evidence supports the telomere-specific DDR that defines mammalian telomere deprotection resulting from macromolecular changes in telomere structure[3–5]. Diverse eukaryotic species arrange their telomeres into the lariat t-loop configuration, where the terminal 3'-overhang of the G-rich telomere sequence strand invades the duplex telomere DNA in cis[6–10]. T-loops in somatic mammalian tissues require the TRFH domain of the TRF2 shelterin subunit[11,12]. Partial TRF2 depletion, or expression of TRF2 TRFH domain mutants in a Trf2−/− background, leads to a telomere-specific and ATM-dependent DDR[13,14] that corresponds with a transition from looped to linear telomeres[11]. Pluripotent murine embryonic stem cells (mESCs) devoid of TRF2 retain t-loops and suppress ATM activity at chromosome ends[15,16]. However, deletion of two shelterin genes in mESCs, Trf1 and Trf2, confer linear telomeres and a corresponding ATM-dependent telomere DDR[15]. The collective data are consistent with a model where chromosome termini are sequestered within t-loops to prevent ATM activation by the naturally occurring DNA ends.

[1]Graduate School of Biostudies, Kyoto University, Sakyo, Kyoto, Japan. [2]IFOM-KU Joint Research Laboratory, Graduate School of Medicine, Kyoto University, Sakyo, Kyoto, Japan. [3]Children's Medical Research Institute, University of Sydney, Sydney, NSW, Australia. [4]IFOM ETS, The AIRC Institute of Molecular Oncology, Milan, Italy. [5]These authors contributed equally: Diana Romero-Zamora, Samuel Rogers. ✉e-mail: makoto.hayashi@ifom.eu; tcesare@cmri.org.au

Canonical telomere deprotection, elicited through aging-dependent telomere erosion, progressively activates two independent tumour suppressive programs[2]. Replicative senescence occurs first and is mediated by a telomere-specific DDR that promotes p53-dependent proliferative arrest[4,17]. p53 compromised cells are refractory to this telomere DDR and continue cell division[17,18] until excessive telomere erosion confers end-to-end chromosome fusions and cell death at crisis[5,19]. Cell lethality during crisis is mediated primarily by autophagic death signalled through the innate immune response[20,21]. A minor proportion of crisis cells, however, perish during mitosis in a MAD manner[22]. Similar mitotic death dependent upon mitotic arrest occurs in response to chemotherapeutic mitotic poisons or lethal replication stress[23,24]. These physiological mitotic death events are promoted in part through a poorly understood and non-canonical mechanism we term MAD-telomere deprotection[22,24–26].

MAD-telomere deprotection is a unique phenomenon where telomere-independent stress promotes active telomere deprotection. In addition to potentiating mitotic death[22], MAD-telomere deprotection also promotes p53-dependent proliferative arrest in the following interphase should cells escape mitotic delay[25,27]. Co-localization of telomeres and DDR markers are termed telomere deprotection induced foci (TIF)[28]. When these events occur due to mitotic arrest, we term them MAD-TIF. MAD-TIF arise in human cells following four or more hours of mitotic arrest and increase proportionally to the time spent in mitotic delay[25]. Diverse genetic or pharmacological interventions that arrest mitosis all confer MAD-TIF[22,24,25], indicating that MAD-telomere deprotection is a physiological response to general mitotic delay. Additionally, MAD-TIF occur irrespective of telomere length and arise independent of telomerase activity[25]. The DDR observed with MAD-TIF is ATM-dependent, suppressed by TRF2 over-expression, and correlates with a transition from looped to linear telomeres[11,25,29]. This suggests that MAD-TIF result from t-loop opening during mitotic arrest. Congruent with a regulated phenomenon, MAD-TIF require AURKB activity but not the mitotic kinases Aurora A or MPS1[25].

MAD-telomere deprotection presents a unique opportunity to explore critical open questions in telomere protection. Foremost is, does shelterin participate in the active process of MAD-telomere deprotection? If so, this changes the conceptual paradigm of shelterin from a solely protective complex to a dynamic entity capable of transitioning telomeres between protected and deprotected states. Furthermore, shelterin components in somatic cells are largely considered to regulate individual DDR activities, with TRF2 being the sole mediator of ATM suppression[1,30]. If additional shelterin components regulate ATM-dependent MAD-TIF, this would represent an unexpected expansion of shelterin protective activities. Additionally, while t-loop junctions are typically presented as a 3-way displacement loop (D-loop), the exact molecular structure at the t-loop insertion point remains conjecture. Alternative configurations include a 4-way Holliday junction (HJ), or a double Holliday junction (dHJ) observed during homologous recombination (HR). The N-terminal TRF2 basic domain binds three- and four-way DNA structures in a sequence-independent manner and is proposed to protect t-loop junctions[31–34]. It remains unclear if the telomere protection afforded by the TRF2 basic domain is ubiquitous or cell-cycle specific. Finally, how t-loop junctions are resolved during mitosis, and if this occurs spontaneously or through recruitment of an independent factor, also remains unknown.

Here we present the development of an unbiased telomere interactomics tool and its application to identify regulators of MAD-telomere deprotection. This revealed that the CPC and BTR complexes associate with TRF1 specifically during mitotic arrest. The CPC is a key mitotic regulator comprised of AURKB, INCENP, Survivin (BIRC5), and Borealin (CDCA8)[35], while BTR promotes dHJ dissolution[36]. We found that AURKB phosphorylated both the TRF1 hinge and the TRF2 basic domains during mitotic arrest and that these modifications were requisite for BTR-dependent MAD-telomere deprotection. The data

reveal unexpected regulation of ATM-dependent MAD-TIF by TRF1 and demonstrate that the TRF2 basic domain protects t-loops from BTR-dependent dHJ dissolution specifically during mitosis. Collectively, the data support a model where coordinated post-translational shelterin modifications promote t-loop unwinding during mitotic arrest to remove damaged cells from the cycling population.

## Results

### TRF1 interacts with the CPC and BLM during mitotic arrest

To identify proteins that interact with human telomeres during mitotic arrest we utilized an unbiased and time-resolved proximity biotinylation strategy. When activated with biotin-phenol and hydrogen peroxide, engineered ascorbic acid peroxidase (APEX2) promiscuously labels proteins within a 10–20 nm radius in 60 s[37] (Supplementary Fig. 1a). We created tri-cistronic lentivectors harboring *mCherry:P2A:-PuroR:T2A:FLAG-APEX2* or *FLAG-APEX2-TRF1* and transduced HeLa cultures. Cells were selected for Puromycin resistance and sorted on mCherry expression to equalize FLAG-APEX2 and FLAG-APEX2-TRF1 expression between conditions (Fig. 1a). We verified APEX2 activity after activation through streptavidin staining of whole cell extracts (Supplementary Fig. 1b). Cytological assessment of streptavidin staining in transduced cultures following one minute of APEX2 activation revealed diffuse localization in Flag-APEX2 cells and spatial enrichment at the telomeres in Flag-APEX2-TRF1 cultures (Fig. 1b). This was indicative of tight spatiotemporal Flag-APEX2-TRF1 labelling, allowing for unbiased time-resolved telomere interactomics.

To enrich for mitotically arrested cells, we synchronized Flag-APEX2 and Flag-APEX2-TRF1 expressing HeLa cultures in G1/S using a single 2 mM thymidine block for 24 h. HeLa cells enter mitosis 6 hours after thymidine washout[11]. We released G1/S synchronized cultures into media containing 150 ng mL$^{-1}$ of the microtubule poison Nocodazole to arrest cells in the subsequent mitosis. APEX2 or APEX2-TRF1 were activated 12, 14, or 16 h after release from G1/S, corresponding with 6, 8, and 10 h of mitotic arrest (Fig. 1c). Following APEX2 or APEX2-TRF1 activation for one minute, we collected samples and recovered biotinylated proteins from cell lysates via streptavidin pulldown (Supplementary Fig. 1c). For these experiments, Flag-APEX2 samples provide reference for non-specific interacting factors, while G1/S synchronized (0-hour post-release) Flag-APEX2-TRF1 samples provide reference for proteins that ubiquitously associate with telomeres.

Streptavidin recovered material was analysed via LC-MS/MS. We refined 7,630 total identifications to 1400 significant proteins, of which 103 were deemed potential Flag-APEX2-TRF1 specific interactors with a Log$_2$Fold-Change > 1 at 12, 14, and, 16 h post-release (Supplementary Fig. 1d). The shelterin components TRF1 (TERF1) and TIN2 (TINF2) were identified as higher confidence interactors by more stringent statistical testing (Fig. 1d, and Supplementary Fig. 1e). Recovery of TRF2 (TERF2) and other shelterin components by Flag-APEX2-TRF1 modestly decreased in abundance during mitotic arrest (Supplementary Fig. 1e), consistent with prior observations of reduced TRF2 at mitotic telomeres during MAD-telomere deprotection[25,26].

Pathway analysis of the 103 potential Flag-APEX2-TRF1 interactors identified an enrichment of mitotic regulators (Supplementary Fig. 1f). This included the kinetochore localized proteins CASC5, SPDL1, CENPE, DSN1, and SGOL2[38]. Additionally, the CPC components INCENP, and Borealin (CDCA8), were strongly enriched within the Flag-APEX2-TRF1 samples (Fig. 1d and Supplementary Fig. 1g). INCENP and Borealin, and less prominently AURKB, follow a similar trend in recovery relative to cell synchrony; all were largely absent at G1/S, 0 h after release from the Thymidine block, and peaked at 12 h post-release corresponding to early MAD-telomere deprotection (Supplementary Fig. 1g). HT1080 6TG cells readily display MAD-telomere deprotection[24]. Co-immunoprecipitations (co-IP) of endogenous TRF1 from HT1080 6TG cells enriched for mitotic arrest

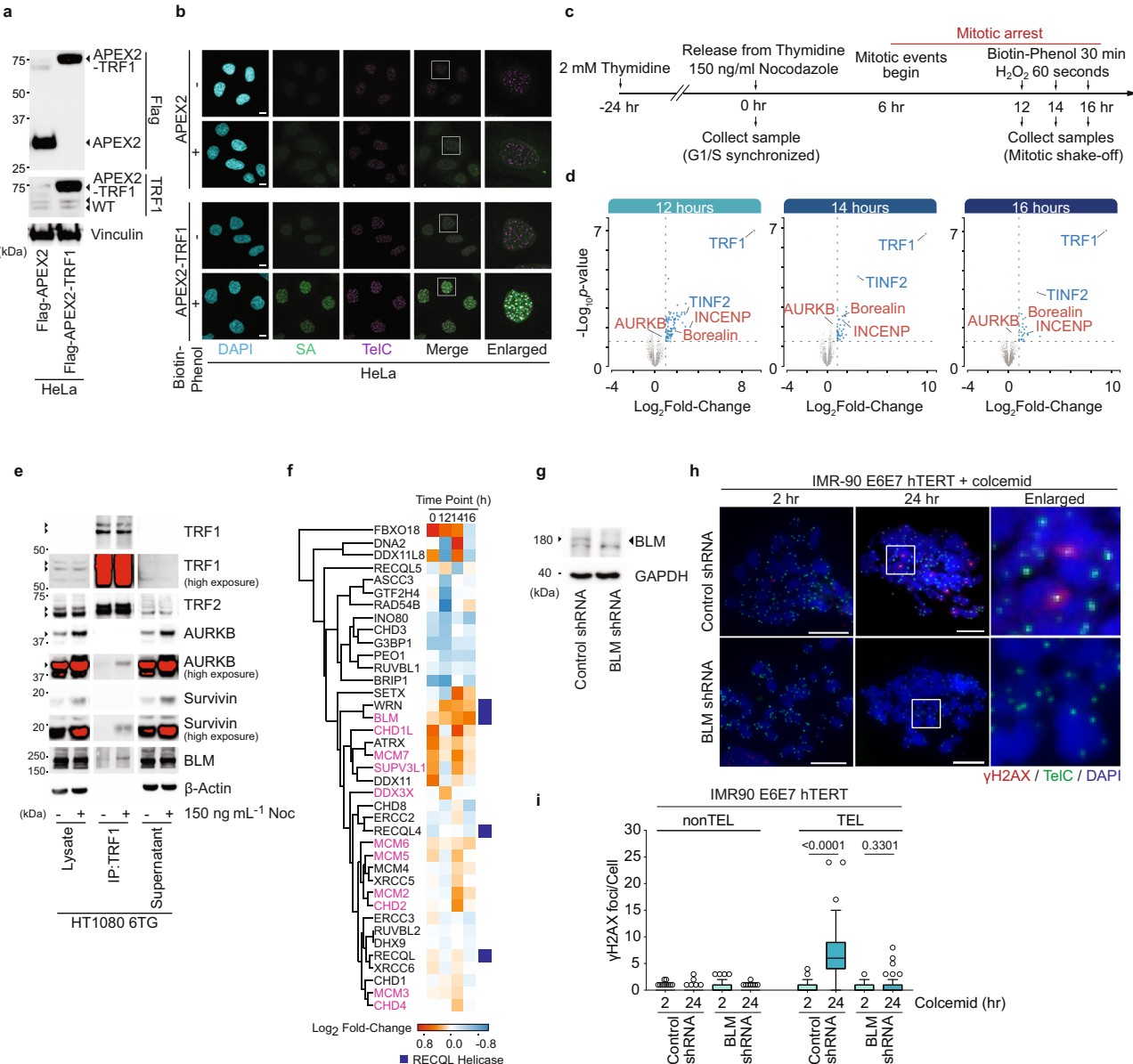

**Fig. 1 | TRF1 interactomics reveal CPC and BLM function in MAD telomere deprotection. a** Immunoblots of whole cell extracts from Flag-APEX2 or Flag-APEX2-TRF1 expressing HeLa (representative of $n = 3$ biological replicates). **b** Combined telomere fluorescence in situ hybridization (FISH, TelC) and streptavidin (SA) labelling in Flag-APEX2 or Flag-APEX2-TRF1 expressing HeLa, ± APEX2 activation by biotin-phenol (representative of $n = 3$ biological replicates). Scale bar, 10 μm. **c** Interactomics timeline. **d** Volcano plots from the Flag-APEX2-TRF1 interactomics described in (**c**) and Supplementary Fig. 1c, d. Proteins enriched from Flag-APEX2-TRF1 samples were compared to Flag-APEX2 at respective timepoints and plotted as $\text{Log}_2$ fold-change and $-\text{Log}_{10}$ $p$-value. Enriched CPC (red) and Shelterin (blue) components are indicated (one-sided student's T-test, $n = 5$ biological replicates). **e** Immunoblots of endogenous TRF1 immuno-precipitates from HT1080 6TG. Where indicated, cells were treated for 16 h with 150 ng mL$^{-1}$ Nocodazole (Noc) immediately prior to sample collection (representative of $n = 3$ biological replicates). **f** Hierarchical clustering of proteins with DNA helicase activity

(GO:0003678) detected by Flag-APEX2-TRF1 interactomics. Colours indicate $\text{Log}_2$-fold change comparing APEX2-TRF1 and APEX2 samples. Helicases with mitotic enrichment are indicated by magenta and RECQL helicases with a blue box. **g** Immunoblots of whole cell extracts from IMR90 E6E7 hTERT expressing Control or BLM shRNAs 5 days post shRNA transduction (representative of $n = 3$ biological replicates). **h** Metaphase-telomere deprotection induced foci (TIF) assays using combined γH2AX immunofluorescence (red) and telomere FISH (TelC, green) in Control or BLM shRNA IMR90 E6E7 hTERT. Cells were treated with 100 ng mL$^{-1}$ colcemid for the indicated time before sample collection (representative from $n = 3$ biological replicates). Scale bar, 10 μm, the DNA is stained with DAPI (blue). **i** Non-telomeric (nonTEL) DNA damage foci and metaphase-TIF (TEL) in shRNA transduced IMR90 E6E7 hTERT treated with 2 or 24 h of 100 ng mL$^{-1}$ colcemid (all data points from $n = 3$ biological replicates of 15 or 30 metaphases per replicate for 2 h and 24 h colcemid, respectively, combined in a Tukey box plot, unpaired two-tailed Mann-Whitney U test). Source data are provided as a Source Data file.

through treatment with 150 ng mL$^{-1}$ Nocodazole for 16 h also recovered AURKB and the additional CPC component Survivin (BIRC5) (Fig. 1e). Similar TRF1 and CPC interactions were not observed in asynchronous controls (Fig. 1e). TRF1 thus interacts with mitotic regulatory proteins critical for MAD-telomere deprotection[25], specifically during mitotic arrest, in multiple human cell lines.

MAD-TIF correlate with a transition from looped to linear telomeres[11], and t-loop unwinding is postulated to require helicase activity[39]. Analysis of our interactomics dataset revealed 39 proteins with DNA helicase activity (GO:0003678), 11 of which were significantly elevated at the telomeres during mitotic arrest (Fig. 1f; highlighted in magenta). Among these helicases, BLM is a known TRF1 interactor[40]

that was proposed to promote dissolution of telomere dHJs should they result from t-loop branch migration[32]. BLM was strongly co-immunoprecipitated with endogenous TRF1 in mitotically arrested but not asynchronous cultures (Fig. 1e). This piqued our interest in BLM as a potential regulator of MAD-telomere deprotection.

MAD-TIF are typically studied in human primary diploid IMR90 fibroblasts[25]. Here we used hTERT immortalized IMR90 that also express papillomavirus serotype 16 E6 and E7 (IMR90 E6E7 hTERT) to respectively inhibit p53 and RB. E6E7 expression facilitates cell division when shelterin proteins are targeted[29]. To measure metaphase-TIF, cells are cytocentrifuged onto glass coverslips and stained with telomere fluorescent in situ hybridization (FISH) and immunoflourenece against γH2AX[41]. MAD-TIF arise following four hours of mitotic delay and increase in number proportional to time spent in mitotic arrest[25]. IMR90 E6E7 hTERT were treated for 2 or 24 h with 100 ng mL$^{-1}$ of the microtubule poison colcemid to arrest mitosis (Fig. 1g–i). With endogenous shelterin, colcemid treatment for 2 h is insufficient to induce MAD-telomere deprotection[25]. Under these conditions, metaphase-TIF present after 2 h of colcemid represent TIF inherited from the prior interphase[29] (Fig. 1i; TEL, Control shRNA). The difference in metaphase-TIF between the 2 and 24 h colcemid treatments represents the MAD-TIF induced through mitotic delay (Fig. 1i; TEL, Control shRNA). We found that BLM shRNA depletion robustly suppressed MAD-TIF in the 24 hr colcemid treated cells (Fig. 1g–i). We observed no increase in non-telomeric mitotic γH2AX foci under any condition (Fig. 1i; non-TEL). This is consistent with prior results showing that mitotic arrest confers a telomere-specific DDR[25]. TRF1 therefore interacts with both AURKB and BLM during mitotic arrest, and BLM promotes MAD telomere deprotection.

## AURKB phosphorylates TRF1 to promote MAD telomere deprotection

The above results inferred a previously unidentified role for TRF1 in MAD-telomere deprotection. To explore further, we shRNA depleted endogenous TRF1 in IMR90 E6E7 hTERT and complemented with exogenous shRNA resistant TRF1 alleles (TRF1$^{shR}$) (Fig. 2a, b, and Supplementary Fig. 2a–e). To our surprise, TRF1 depletion suppressed MAD-TIF in cultures treated with colcemid for 24 h (Fig. 2b and Supplementary Fig. 2a). MAD-TIF were rescued with TRF1$^{shR}$-wild type (WT) but not TRF1$^{shR}$-ΔBLM which carries a deletion of the BLM-binding motif[40] (Fig. 2b and Supplementary Fig. 2b, c).

MAD-telomere deprotection requires AURKB activity[25]. In silico analysis of the TRF1 protein sequence near the BLM-binding motif identified three potential $(R/K)_{1-3}$-X-S/T AURKB phosphorylation sites[42], Ser296, Ser354, and Thr358. Ectopic expression of TRF1$^{shR}$ alleles carrying an Ala mutation of all or each residue revealed that Ser354 and Thr358 were required for MAD-TIF formation (Fig. 2b, and Supplementary Fig. 2c, d). Mutation of both S354 and T358 to phospho-mimetic Asp (TRF1$^{shR}$-2D) restored MAD-TIF (Fig. 2c, and Supplementary Fig. 2e), suggestive of TRF1 phospho-regulation being requisite for MAD-telomere deprotection. TRF1$^{shR}$-T358A, but not TRF1$^{shR}$-WT, failed to rescue the TRF1 shRNA phenotype in carcinoma cell line HCT116 and fibrosarcoma cell line HT1080, confirming that this function is not cell-type specific (Supplementary Fig. 2f). We note that S354, and R355/R356 of the RRxT motif for T358, are absent from the TRF1$^{shR}$-ΔBLM allele that fails to rescue MAD-TIF. To determine if mutating the potential AURKB phospho-sites affects mitotic TRF1 localization, we tagged TRF1 variants with mScarlet and co-expressed with mClover-TRF2 (Supplementary Fig. 2g). We found that both mScarlet-TRF1$^{shR}$-3A and mScarlet-TRF1$^{shR}$-2D maintained their telomere localization in the 24 h colcemid treated cells (Supplementary Fig. 2g).

We attempted to generate phospho-specific antibodies against these potential TRF1 phospho-sites. The phospho-TRF1-Ser354 (pTRF1-S354) antigen failed to produce specific antibody titres. Phospho-TRF1-

Thr358 (pTRF1-T358), however, produced an antibody that reliably detected a band of approximately 60 kDa, corresponding to the size of Flag-TRF1 (Supplementary Fig. 2h). The band was detected in anti-Flag immuno-precipitates from shTRF1 and Flag-TRF1$^{shR}$ expressing HT1080 6TG cells enriched for mitotic arrest through G1/S synchrony and release into nocodazole for 16 h (Supplementary Fig. 2h). The band was absent in Flag-TRF1$^{shR}$ immuno-precipitates from cultures without nocodazole, Flag-TRF1$^{shR}$-T358A immuno-precipitates under all experimental conditions, and in alkaline phosphatase treated immuno-precipitates from Flag-TRF1$^{shR}$ cells enriched for mitotic arrest (Supplementary Fig. 2h). These data confirm antibody specificity and reveal accumulation of pTRF1-T358 during mitotic arrest.

To explore potential AURKB regulation of pTRF1-T358, HT1080 6TG cultures were G1/S synchronized, released, and enriched for mitotic arrest with 16 h of 150 ng mL$^{-1}$ Nocodazole in the presence or absence of mitotic kinase inhibitors (Fig. 2d). The AURKB inhibitor Hesperadin was previously demonstrated to suppress MAD-TIF formation at 40 nM concentration[22]. Under these mitotic arrest conditions, 40 nM Hesperadin prevented pTRF1-T358 from exceeding interphase levels, while inhibitors against BUB1 (BAY1816032)[43] or CDC7 (XL413)[44] produced no effect (Fig. 2d). Additionally, low concentrations of AURKB vigorously phosphorylated purified Flag-TRF1 in vitro as measured by the anti-pTRF1-T358 antibody (Fig. 2e). Collectively the data indicate that TRF1 is required for MAD-telomere deprotection. Further, the data demonstrate that the role of TRF1 in this phenomenon is regulated through AURKB phosphorylation of TRF1 at T358 and potentially S354.

## Phosphorylated TRF1 recruits the CPC through survivin

The TRF1 hinge region interacts with multiple proteins including BLM[40]. To identify proteins that interact with the AURKB modified TRF1 residues, we chemically synthesised phosphopeptides corresponding to pTRF1-S354 and pTRF1-T358. pTRF1-S296, shown above to not participate in MAD-telomere deprotection, and the known ATM site pTRF1-S367[45] were included as controls. The phosphopeptides, and their respective non-phosphorylated controls, were immobilised on streptavidin beads and incubated with lysates from HT1080 6TG cultures enriched for mitotic arrest through G1/S synchrony and release for 16 h in nocodazole (Fig. 3a, b). LC-MS/MS of proteins recovered by TRF1 peptide pulldown did not show enrichment of BLM nor its binding partners. Instead, we identified enrichment of the CPC component Survivin in both the pTRF1-S354 and pTRF1-T358 peptide samples relative to non-phosphorylated controls (Fig. 3c and Supplementary Fig. 3a). Other CPC members Borealin and AURKB were only weakly detected across all peptide pulldowns. Within the pTRF1-S354 sample, we also recovered the 14-3-3 complex that is known to bind promiscuously to phosphorylated peptides[46] (Supplementary Fig. 3a).

To evaluate in a cellular context, we performed Flag co-IP in lysates from HT1080 6TG cells expressing TRF1$^{shR}$-WT or mutant alleles (Fig. 3d). Cells were enriched for mitotic arrest through G1/S synchrony and released into nocodazole for 16 h before lysate collection. Interaction between TRF1 and the CPC components Survivin, INCENP, and AURKB was abrogated by deletion of the TRF1 hinge region (Flag-TRF1$^{shR}$-ΔHD, aa336-367) that includes the S354 and T358 residues, and with double phospho-null Ala substitutions on S354 and T358 (Flag-TRF1$^{shR}$-2A) (Fig. 3d). CPC recovery was partially rescued by Flag-TRF1$^{shR}$-2D carrying dual phospho-mimetic Asp substations at S354 and T358 (Fig. 3d). Recovery of BLM and its binding partner TOP3A were also reduced in Flag precipitates from Flag-TRF1$^{shR}$-ΔHD and Flag-TRF1$^{shR}$-2A expressing cells and did not display obvious rescue in samples from Flag-TRF1$^{shR}$-2D cells (Fig. 3d). pTRF1-S354 and pTRF1-T358 thus constitute a novel Survivin binding site.

To address if CPC components other than AURKB function in MAD-telomere deprotection, we shRNA depleted INCENP or Survivin in IMR90 E6E7 hTERT (Supplementary Fig. 3b). Depleting CPC

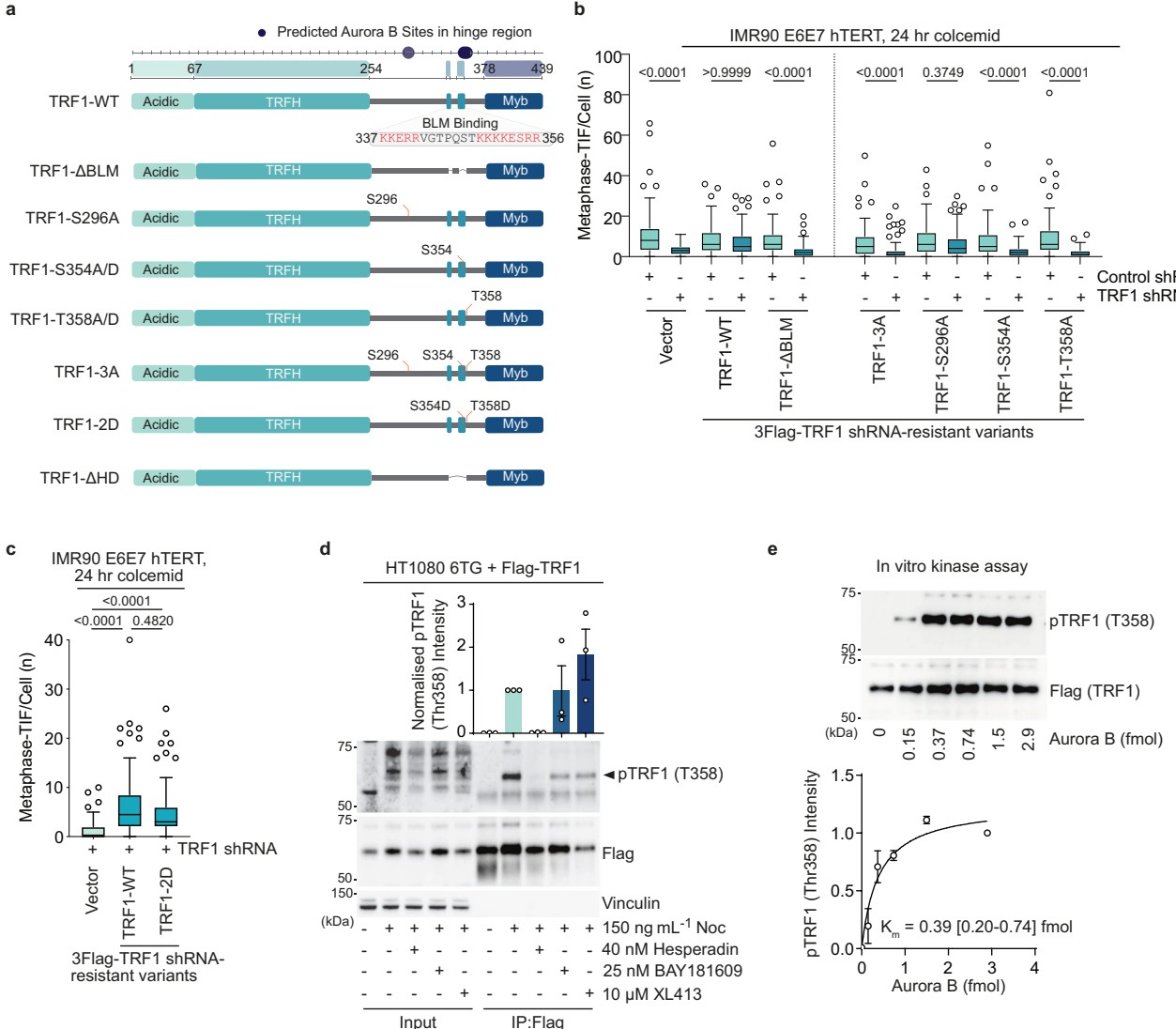

**Fig. 2 | MAD telomere deprotection requires TRF1 phosphorylation by AURKB.**
**a** TRF1 domain structure and the mutant alleles used in this study. Predicted AURKB sites in the hinge region and BLM binding domain are shown. The deleted amino acids within the FLAG-TRF1$^{\Delta BLM}$ variant are indicated in red. **b**, **c** Metaphase-telomere deprotection induced foci (TIF) following 24 h of 100 ng mL$^{-1}$ colcemid in IMR90 E6E7 hTERT expressing Control or TRF1 shRNA and vector or shRNA-resistant TRF1 alleles (all data points from $n = 3$ biological replicates of 30 metaphases per replicate combined into a Tukey boxplot, Kruskal-Wallis followed by Dunn's multiple comparisons test). **d** Below, representative immunoblots of Flag immuno-precipitates from TRF1 shRNA HT1080 6TG cells expressing shRNA-resistant WT Flag-TRF1. Cells were synchronized with a thymidine block, released, and treated with 150 ng mL$^{-1}$ nocodazole (Noc) where indicated, in the presence or absence of indicated kinase inhibitors for 16 h. The pTRF1-T358 band is indicated with an arrow. Above, quantitation of normalized anti-TRF1-pT358 immunoblot signal (mean +/- s.e.m., $n = 3$ biological replicates). **e** Above, example of an anti-TRF1-pT358 immunoblot measuring in vitro AURKB kinase assay on purified Flag-TRF1. Below, quantitation of anti-TRF1-pT358 intensity and the resulting Km (mean +/- s.e.m., $n = 3$ biological replicates, $K_m$ is 0.39 (95% CI: 0.2–0.74) fmol). Source data are provided as a Source Data file.

components could affect mitotic arrest[47]. We therefore performed live imaging on 100 ng ml$^{-1}$ colcemid treated cultures, with or without 40 nM Hesperadin, INCENP, or Survivin shRNA. We found that mitotic duration at 24 h colcemid in 40 nM Hesperadin treated or CPC depleted cells was consistent with mitotic duration in CPC functional cells treated with 16 h colcemid (Supplementary Fig. 3c, d). We compared metaphase-TIF in these conditions of equal mitotic arrest which revealed that *INCENP*, *Survivin*, and AURKB activity are all required for MAD-TIF (Supplementary Fig. 3e).

Collectively the data demonstrate that AURKB phosphorylates the TRF1 hinge domain to promote TRF1 and CPC interaction via Survivin. Consistent with CPC involvement in mitotic telomere deprotection, multiple CPC subunits are required for MAD-TIF. We note the BLM and Surivin interaction motifs overlap on TRF1, indicative of possible

competition for binding during mitotic arrest. Survivin binding to AURKB-modified TRF1 suggests functional significance for CPC retention on Shelterin during mitotic arrest, potentially leading to modification of other telomere proteins.

## AURKB phosphorylates the TRF2 basic domain to promote MAD telomere deprotection
MAD-TIF are ATM-dependent, suppressed by TRF2 over-expression, and enhanced by partial TRF2 depletion[25,26]; indicative of a central role for TRF2 in phenomenon regulation. We thus postulated AURKB may also modify TRF2. In silico analysis identified Ser62 and Ser65 in the human TRF2 basic domain as potential AURKB consensus sites (Fig. 4a). This was intriguing as the N-terminal TRF2 basic domain was previously suggested to protect t-loop junctions[32,33].

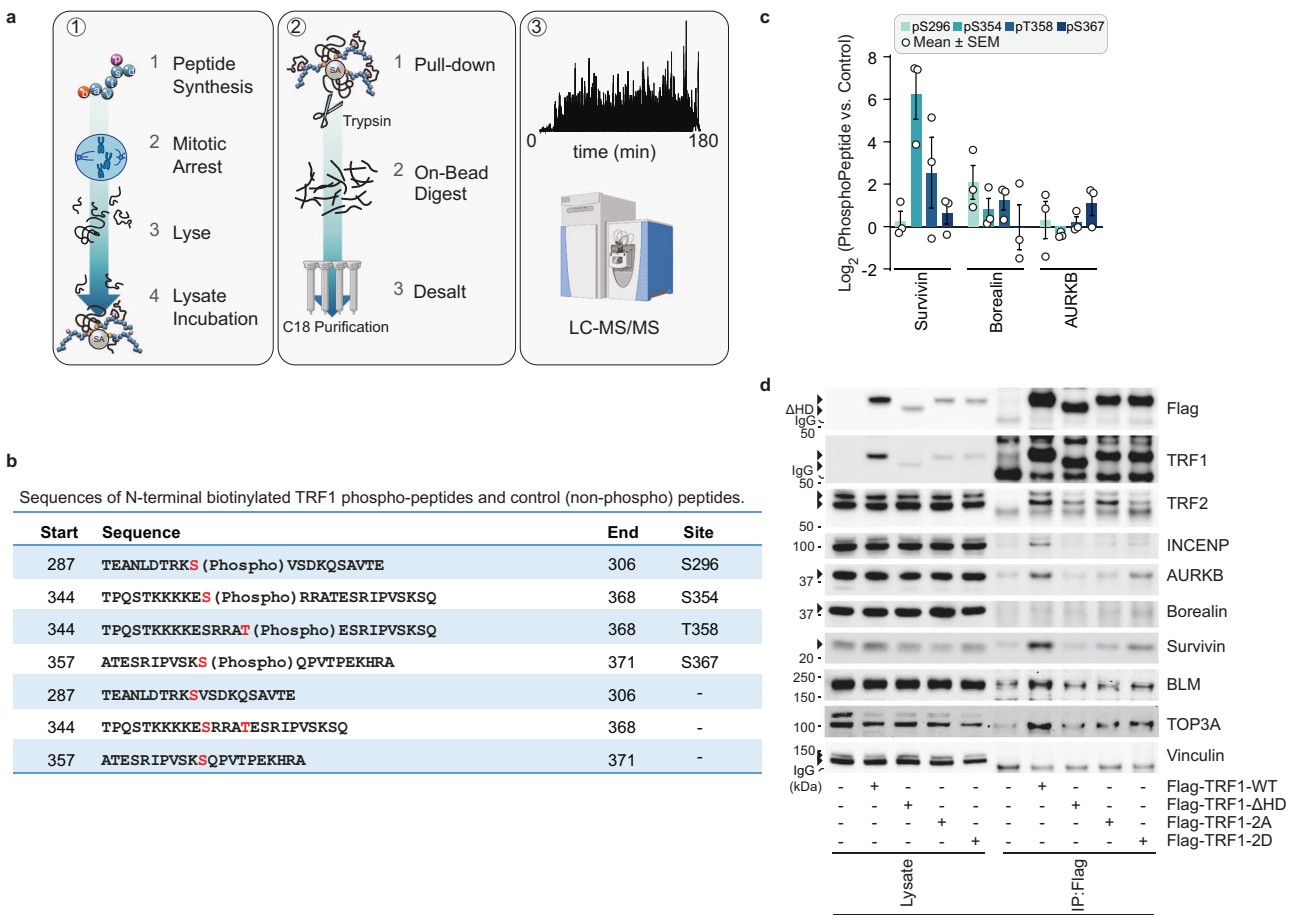

**Fig. 3 | Survivin binds to TRF1 phosphorylated on Ser354 and Thr358.**
**a** Schematic of TRF1 peptide pull-down and subsequent LC-MS/MS analysis for the experiment in (**b**, **c**) and Supplementary Fig. 3a. To enrich for mitotic arrest, HT1080 6TG cells were synchronised with a thymidine block and released in the presence of 150 ng mL$^{-1}$ Nocodazole for 16 h before sample collection and extract preparation. TRF1 peptides immobilized onto beads were incubated with HT1080 6TG mitotic extracts before LC-MS/MS sample preparation and analysis. Drawing of the mass spectrometer was created in BioRender (Cesare, T. (2025) https://BioRender.com/r86h834). **b** Summary of peptides used for pull-down analysis in (**c**)

and Supplementary Fig. 3a. Red indicates S296, S354, T358, and S367. **c** Log$_2$-fold change in Chromosome Passenger Complex proteins recovered from mitotically arrested HT1080 6TG extracts by TRF1 phosphopeptides (mean +/- s.e.m., $n = 3$ biological replicates). **d** Immunoblots of anti-Flag immuno-precipitates from TRF1 shRNA HT1080 6TG expressing the indicated shRNA-resistant Flag-TRF1-WT or mutant alleles. Cells were synchronised with a thymidine block, released, and treated with 150 ng mL$^{-1}$ of nocodazole for 16 h. TRF1-DHD is a deletion of residues 336–367 (representative of $n = 3$ biological replicates is shown). Source data are provided as a Source Data file.

We attempted to generate phospho-specific antibodies and failed for phospho-TRF2-Ser62 (pTRF2-S62). Anti-phospho-TRF2-Ser65 (pTRF2-S65), however, returned positive data. For these experiments, we depleted endogenous TRF2 via shRNA and expressed exogenous TRF2-WT or mutant alleles. In whole cells extracts from HT1080 6TG cultures enriched for mitotic arrest with G1/S synchrony and released into nocodazole for 16 hours (Supplementary Fig. 4a, Input), pTRF2-S65 revealed a band of approximately 60 kDa in Myc-TRF2-WT but not Myc-TRF2-S65A expressing cells. (Supplementary Fig. 4a, Input). Immunoprecipitation of Myc-TRF2 from mitotically arrested cells revealed an enhanced band of the same molecular weight (Supplementary Fig. 4a, IP:myc). We note a band of the same molecular weight, but less intense, was detectable following IP of Myc-TRF2-S65A from interphase and mitotically arrested cultures. We anticipate the pTRF2-S65 antibody may weakly recognize other phospho-sites on purified myc-TRF2, potentially Ser62, when Ser65 is no longer available for modification (Supplementary Fig. 4a, IP:myc).

Consistent with pTRF2-S65 phosphorylation via AURKB, Hesperadin pre-treatment of cultures enriched for mitotic arrest reduced the intensity of the pTRF2-S65 band from Myc-TRF2 immunoprecipitates

52.1 ± 12.7% (mean ± s.d.) (Fig. 4b). TRF2 phosphorylation was unaffected by treatment with the Bub1 or CDC7 inhibitors BAY1816032 or XL413 (Fig. 4b). Additionally, incubation of immunoprecipitated Myc-TRF2 with recombinant AURKB in vitro produced substantial TRF2 phosphorylation as detected with the anti-pTRF2-S65 antibody, albeit with slower kinetics than TRF1 phosphorylation (Fig. 4c).

To explore the roles of TRF2-S62 and -S65 in mitotic telomere protection we shRNA depleted endogenous TRF2 in IMR90 E6E7 hTERT and complemented with ectopic wild-type or mutant TRF2 (Fig. 4d). Congruent with prior results[25,29]: TRF2 depletion conferred interphase-TIF and metaphase-TIF with 2 h of 100 ng mL$^{-1}$ colcemid[29], consistent with passage of interphase-TIF into mitosis (Fig. 4d, e and Supplementary Fig. 4b-e); TRF2 depletion also exacerbated the number of metaphase-TIF observed with 24 as compared to 2 h of colcemid, indicative of an enhanced MAD-TIF response[29] (Fig. 4e, Supplementary Fig. 4e); and overexpression of ectopic TRF2-WT in a TRF2 shRNA background suppressed interphase-TIF and metaphase-TIF under all colcemid conditions[25,29] (Fig. 4d, e and Supplementary Fig. 4c–e). Notably, all TRF2-S62 and/or 65 mutant alleles rescued interphase protection in a TRF2 shRNA background (Fig. 4d and Supplementary Fig. 4c, d, f).

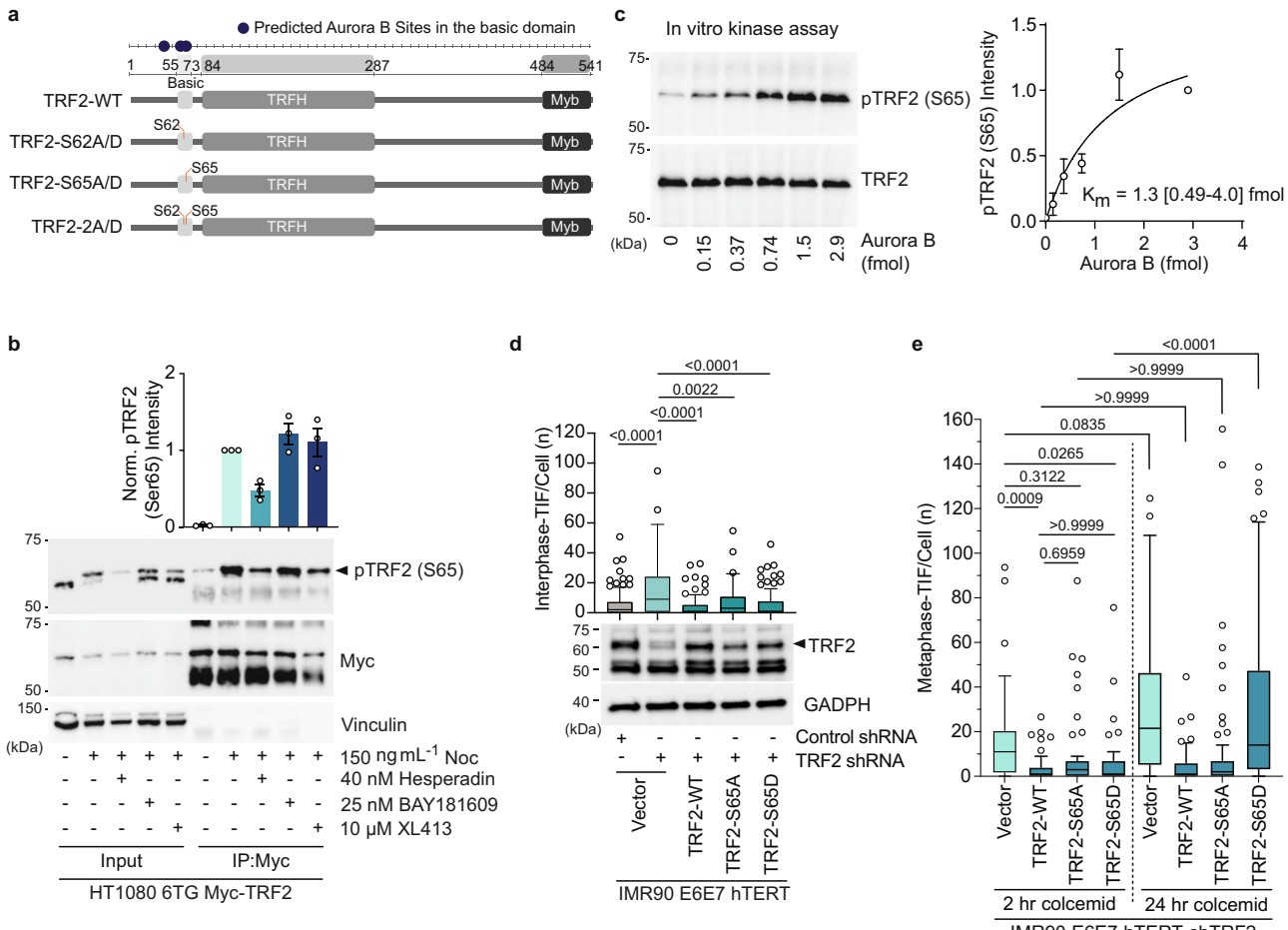

**Fig. 4 | AURKB phosphorylates the TRF2 basic domain to promote MAD telomere deprotection. a** TRF2 domain structure and the mutant alleles used in this study. Predicted AURKB sites in the basic domain are indicated. **b** Below, representative immunoblots of Myc immuno-precipitates from TRF2 shRNA HT1080 6TG expressing Myc-TRF2-WT. Cells were synchronized with a thymidine block, released in the presence or absence of 150 ng mL⁻¹ nocodazole (Noc) and the indicated kinase inhibitors for 16 h. Above, quantitation of normalized anti-TRF2-pS65 immunoblot signal (mean +/- s.e.m., *n* = 3 biological replicates). **c** Left, example of anti-TRF2-pS65 immunoblot measuring in vitro AURKB kinase assay on purified Myc-TRF2. Right, quantitation of anti-TRF2-pS65 intensity and the resulting Km (mean +/- s.e.m., *n* = 3 biological replicates, Km is 1.3 (95% CI: 0.49–4.0)

fmol). **d**, **e** Quantitation of interphase-telomere deprotection induced foci (TIF) (**d**) and metaphase-TIF (**e**) in TRF2 shRNA IMR90 E6E7 hTERT expressing TRF2-WT or the indicated TRF2 alleles. For (**d**) Above, all data points from *n* = 3 biological replicates of 45 nuclei per replicate compiled into a Tukey boxplot, Kruskal-Wallis followed by Dunn's multiple comparisons test. Below, representative immunoblots of whole-cell extracts derived from the cell cultures used in this experiment. For (**e**), cells were treated with 2 or 24 h of 100 ng mL⁻¹ colcemid (all data points from *n* = 3 biological replicates of 15 and 30 metaphases per replicate for 2 h and 24 h colcemid, respectively, compiled into a Tukey boxplot, Kruskal-Wallis followed by Dunn's multiple comparisons test). Source data are provided as a Source Data file.

Examination of TRF2-S65 revealed straightforward evidence consistent with phosphorylation of this residue being requisite for MAD-telomere deprotection. With two hours colcemid there was no difference in metaphase-TIF between WT-TRF2, TRF2-S65A and TRF2-S65D (Fig. 4e). Whereas the phospho-mimetic TRF2-S65D, but not phospho-null TRF2-S65A, was permissive to metaphase-TIF with 24 h colcemid (Fig. 4e). Analysis of the TRF2-S62 mutants, however, revealed a contextually subtle phenotype. While all TRF2-S62 mutants, including TRF2-S62 A/S65A (TRF2-2A) and TRF2-S62D/S65D (TRF2-2D), suppressed interphase-TIF, all TRF2-S62 mutations also permitted metaphase-TIF with both two and 24 h of colcemid (Supplementary Fig. 4f, g). This indicated that within the context of TRF2-S62 mutation, metaphase-TIF observed with two hours of colcemid are not interphase-TIF passed into mitosis. Instead, the metaphase-TIF observed with two hours of colcemid in TRF2-S62 mutants represent MAD-TIF that arise through accelerated kinetics. We anticipate the sensitivity of TRF2-S62 to both phospho-null and -mimetic substitutions indicates the central importance of this residue in mitotic telomere protection. Notably, both mScarlet-TRF2-2A and -2D localized to

chromosome ends with 24 hr colcemid indicating that TRF2 mutation did not affect telomere localization (Supplementary Fig. 2g).

We predicted CPC retention on phosphorylated TRF1 may potentiate subsequent TRF2 phosphorylation. In agreement, depleting TRF1 reduced TRF2 phosphorylation in cells treated with 150 ng mL⁻¹ nocodazole for 16 h. (Supplementary Fig. 4h, i). Collectively, the findings implicate the TRF2 basic domain in mitosis-specific telomere protection and reveal this protective capacity is attenuated through TRF1 and AURKB-dependent modification.

**Attenuation of the TRF2 basic domain promotes MAD t-loop opening**

To determine if AURKB modification of the TRF2 basic domain impacts mitotic t-loops, we sought to directly visualize telomere macromolecular structure through near super-resolution Airyscan microscopy[11]. For this purpose, we used *Trf2^Floxed/Floxed Rosa26-CreERT2 pBabeSV40LT* murine embryonic fibroblasts (hereafter abbreviated *Trf2^F/F CreER LgT* MEFs) which contain long telomeres that are easier to resolve with enhanced fluorescence microscopy[12]. *Trf2^F/F CreER LgT* MEFs were complimented

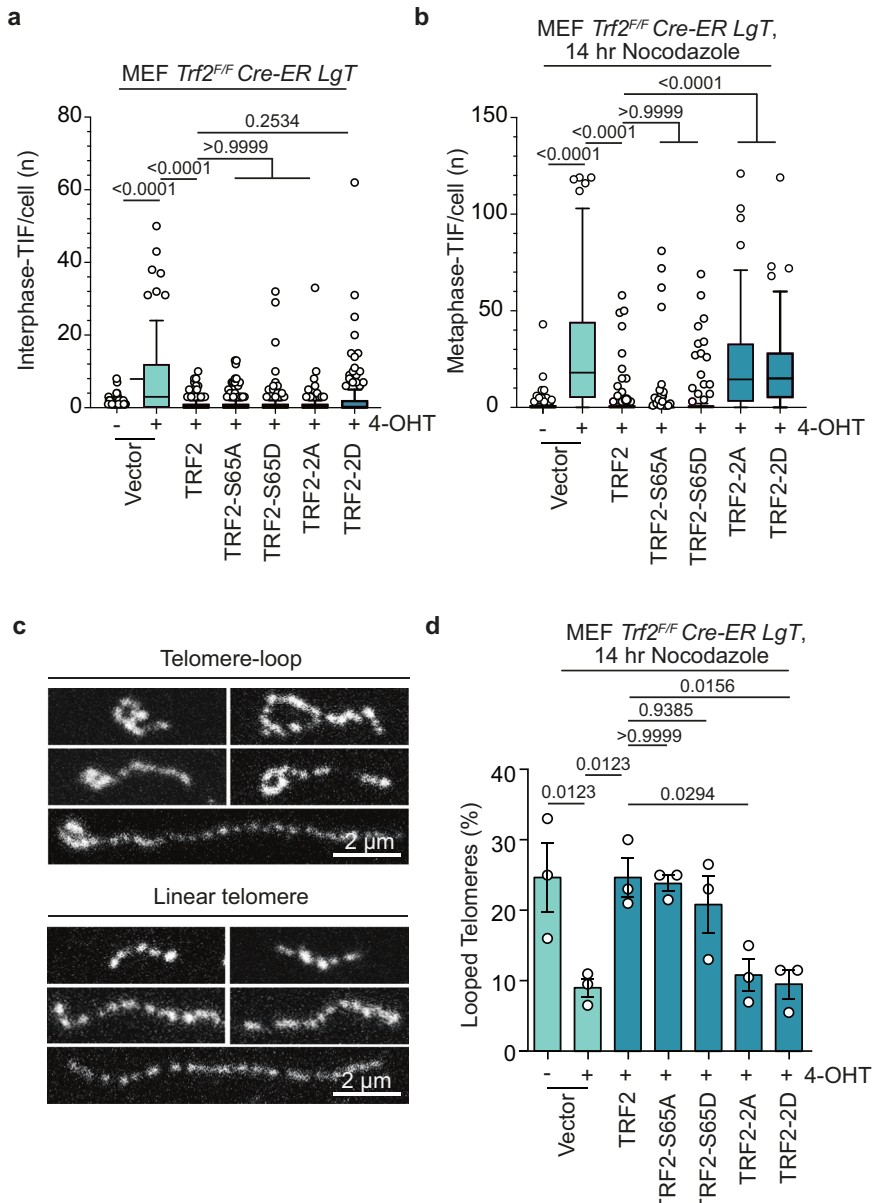

**Fig. 5 | TRF2 basic domain modification during mitotic arrest promotes t-loop opening. a, b** Interphase-telomere deprotection induced foci (TIF) (**a**) and Metaphase-TIF (**b**) in *Trf2^F/F^ Cre-ER LgT* MEFs expressing TRF2-WT or the indicated TRF2 alleles. Where indicated, endogenous *mTrf2* was deleted by 4-Hydroxytamoxifen (4-OHT) addition 36 h prior to sample fixation. In (**b**) cells were treated with 14 h of 400 ng mL⁻¹ Nocodazole prior to sample preparation (all data points from *n* = 3 biological replicates of ≥ 50 cells (**a**) or 30 metaphases (**b**) per replicate, compiled into a Tukey box plot, Kruskal-Wallis followed by Dunn's multiple comparisons test). **c** Representative examples of telomere macromolecular structure as visualized by AiryScan microscopy from *Trf2^F/F^ Cre-ER LgT* MEFs treated with 400 ng mL⁻¹ Nocodazole for 14 h and collected by mitotic shake-off. Samples were trioxsalen cross-linked in situ, and the chromatin spread on coverslips through cytocentrifugation before telomere FISH labelling[11]. Scale bar, 2 μm. Representative of *n* = 3 biological replicates. **d** Quantification of looped telomeres from *Trf2^F/F^ Cre-ER LgT* MEFs expressing the indicated TRF2 alleles in mitotically arrested samples prepared and shown as in (**c**) and Supplementary Fig. 5c. Where indicated, endogenous *mTrf2* was deleted by 4-OHT addition 36 hours prior to sample fixation (mean +/- s.e.m., *n* = 3 biological replicates of 200 telomeres per replicate, Ordinary one-way ANOVA followed by Šídák's multiple comparisons test, F = 6.463, DF = (6, 14)). Source data are provided as a Source Data file.

with retroviral vectors expressing ectopic human TRF2 alleles and selected for transduction (Supplementary Fig. 5a). Following endogenous *mTrf2* deletion with 4-hydroxytamoxifen (4-OHT) we assessed interphase-TIF and metaphase-TIF in cells arrested with Nocodazole for 14 h. Consistent with IMR90 E6E7 hTERT, all hTRF2 mutant alleles restored interphase telomere protection in 4-OHT treated *Trf2^F/F^ CreER LgT* MEFs (Fig. 5a and Supplementary Fig. 5b). Also congruent between IMR90 E6E7 hTERT and 4-OHT treated *Trf2^F/F^ CreER LgT* MEFs, hTRF2-WT and hTRF2-S65A suppressed MAD-TIF while the hTRF2-2A and hTRF2-2D alleles harboring S62 mutations did not (Fig. 5b and Supplementary Fig. 5b). We note a subtle difference between MEFs and IMR90 E6E7 hTERT

hTERT, as hTRF2-S65D was susceptible to MAD-TIF in the human but not the murine context (Figs. 4e, 5b).

To assess mitotic telomere configuration, we 4-OHT treated *Trf2^F/F^ CreER LgT* MEFs and 36 hours later enriched cultures for mitotic arrest with 14 hr of 400 ng mL⁻¹ nocodazole. Mitotic cells were collected via shake off and interstrand DNA crosslinks were introduced in situ with trioxsalen and UV light[6]. Interstrand cross-linking is required to maintain t-loop structure[6,11,12]. The chromatin was then cytocentrifuged onto glass coverslips in the presence of a mild detergent to decompact the telomeres prior to staining with telomere FISH[11,12]. Images were collected by airyscan microscopy and

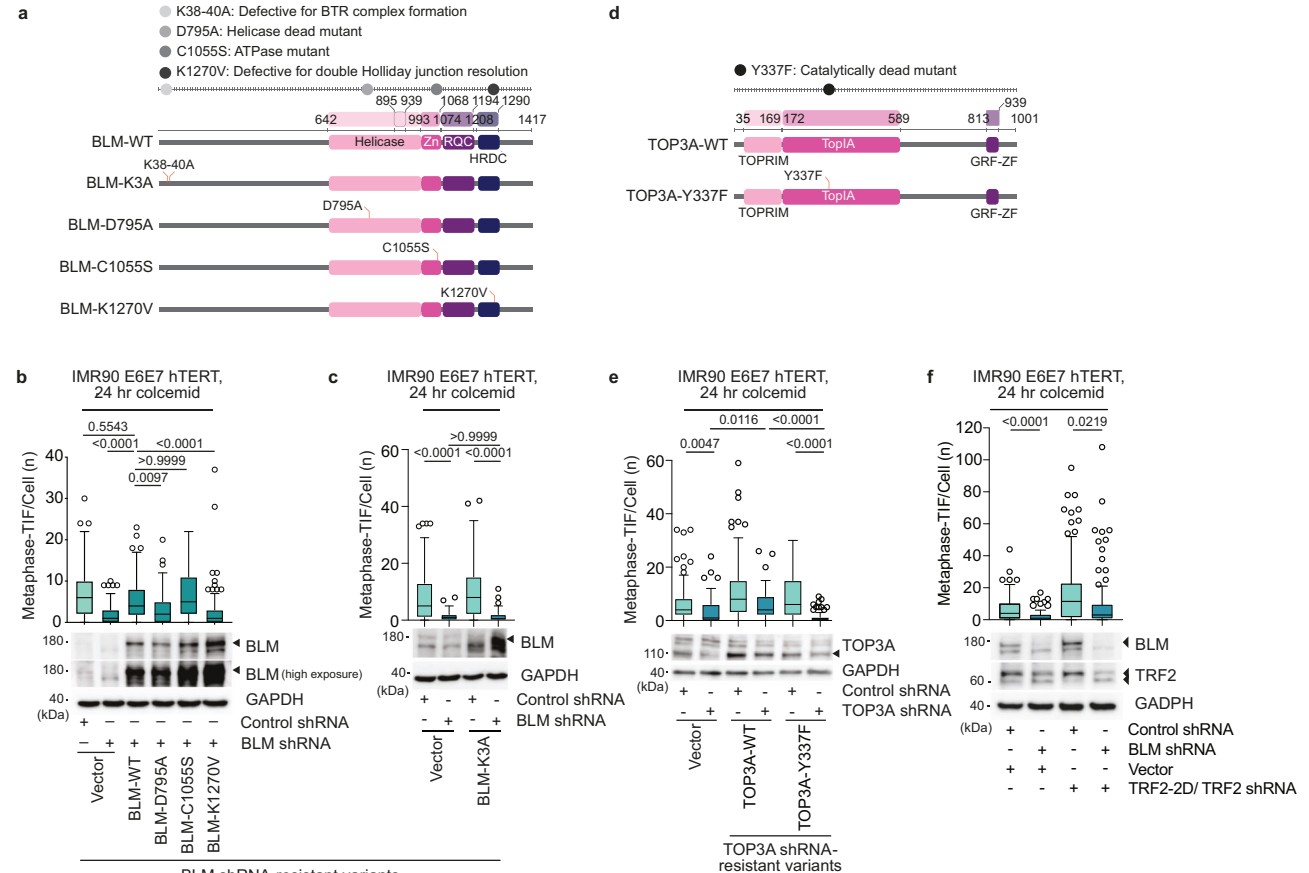

**Fig. 6 | TRF2 phosphorylation promotes BTR-dependent MAD telomere deprotection. a** BLM domain structure and the mutant alleles used in this study. The compromised function of the mutant alleles is indicated. **b, c** Above, Metaphase-telomere deprotection induced foci (TIF) following 24 h of 100 ng mL$^{-1}$ colcemid in IMR90 E6E7 hTERT expressing Control or BLM shRNA and shRNA-resistant BLM alleles (all data points from $n = 3$ biological replicates of 30 metaphases per replicate, compiled into a Tukey Box Plot, Kruskal-Wallis followed by Dunn's multiple comparisons test). Below: representative immunoblots of whole cell extracts derived from the untreated cell cultures used in this experiment. **d** TOP3A domain structure and the mutant allele used in this study. **e** Above, Metaphase-TIF following 24 h of 100 ng mL$^{-1}$ colcemid in IMR90 E6E7 hTERT fibroblasts expressing Control or TOP3A shRNA and vector or shRNA-resistant

TOP3A alleles (all data points from $n = 3$ biological replicates of 30 metaphases per replicate compiled into a Tukey Box Plot, Kruskal-Wallis followed by Dunn's multiple comparisons test). Below, representative immunoblots of whole cell extracts derived from the untreated cell cultures used in this experiment. **f** Above, Metaphase-TIF following 24 h of 100 ng mL$^{-1}$ colcemid in IMR90 E6E7 hTERT expressing Control, BLM or TRF2 shRNA, and vector or TRF2-2D (S62D and S65D) (all data points from $n = 3$ biological replicates of 30 metaphases per replicate compiled into a Tukey Box Plot, Kruskal-Wallis followed by Dunn's multiple comparisons test). Below, representative immunoblots of whole cell extracts derived from the untreated cell cultures used in this experiment. Source data are provided as a Source Data file.

telomeres scored as looped or linear in blinded samples (Fig. 5c and Supplementary Fig. 5c).

Quantitation of telomere configuration in mitotically arrested cells revealed approximately 20% looped mitotic telomeres in *Trf2*^F/F^ *CreER LgT* MEFs with endogenous *Trf2*, and in 4-OHT treated samples expressing hTRF2-WT, hTRF2-S65A, or hTRF2-S65D (Fig. 5d). Twenty-percent looped telomeres is consistent with prior observations of cells containing wholly protected telomeres[11,12,15,39], and likely represents the limitation of trioxsalen crosslinking of t-loop junctions in situ[11]. Conversely, we found a diminished frequency of looped telomeres in 4-OHT treated *Trf2*^F/F^ *CreER LgT* MEFs transduced with vector, hTRF2-2A, or hTRF2-2D (Fig. 5d). The inverse correlation observed between metaphase-TIF and t-loops during mitotic arrest (compare Fig. 5b and d) is consistent with mitotic attenuation of the TRF2 basic domain promoting MAD-TIF through telomere linearization.

**BTR activity is required for MAD telomere deprotection**
The above observations are consistent with the TRF2 basic domain counteracting BLM-dependent MAD-telomere deprotection. To determine which attributes of BLM promote MAD-TIF, we shRNA

depleted BLM in IMR90 E6E7 hTERT and ectopically expressed shRNA resistant BLM alleles (BLM^shR^, Fig. 6a). In cultures treated for 24 h with 100 ng mL$^{-1}$ colcemid, BLM^shR^-WT promoted MAD-TIF as did the BLM^shR^-C1055S ATPase mutant that is mislocalized in interphase cells and defective for DNA duplex unwinding[48] (Fig. 6a, b). Conversely, MAD-TIF were not rescued by mutations in the helicase (BLM^shR^-D795A)[49] and HRDC domains (BLM^shR^-K1270V)[50] that attenuate dHJ dissolution (Fig. 6a, b).

Within the spectrum of BLM activities, dHJ dissolution is carried out by the BTR complex[36]. Expression of the BLM N-terminus Lys38-40 (BLM^shR^-K3A)[51] mutant that disrupts interaction between BLM and other BTR components abolished MAD-TIF formation (Fig. 6c). As did individual depletion of the BTR components *TOP3A*, *RMI1*, or *RMI2* (Supplementary Fig. 6a, b). MAD-TIF were also rescued in TOP3A-depleted cells by TOP3A^shR^-WT but not the enzyme-dead mutant TOP3A^shR^-Y337F[52] (Fig. 6d, e). As a control, we depleted *FANCJ*, a BLM associated helicase that operates outside of BTR[53] (Supplementary Fig. 6a). This had no effect on MAD-TIF (Supplementary Fig. 6b). Together this causal evidence links BTR catalytic activity to MAD-telomere deprotection.

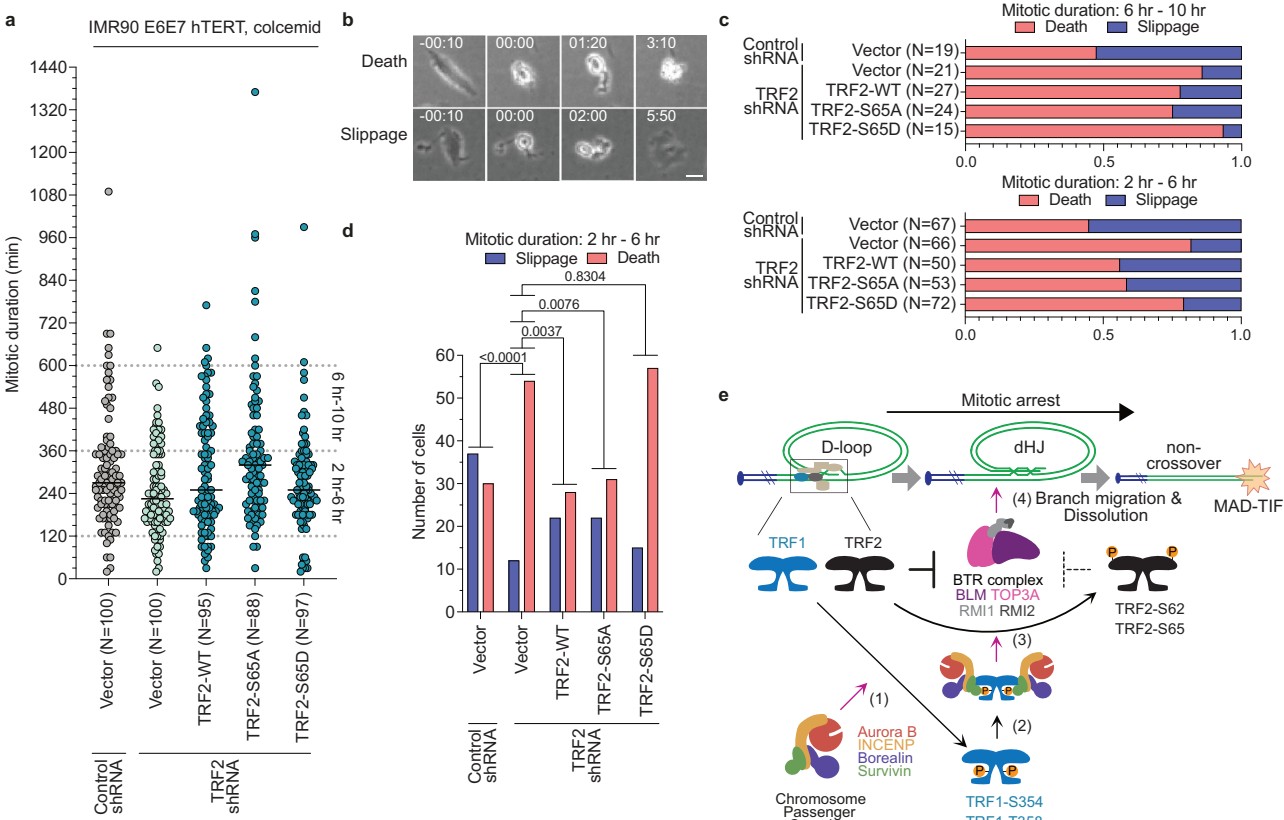

**Fig. 7 | TRF2 phosphorylation promotes mitotic death during mitotic arrest.**
**a** Mitotic duration for TRF2 shRNA IMR90 E6E7 hTERT expressing the indicated TRF2 alleles. Cells were observed for 50 h in cultures treated with 100 ng mL$^{-1}$ colcemid (median of N cells from $n = 2$ experimental replicates). **b** Representative captures from phase contrast live imaging in (**a**). Mitotic outcomes are indicated. Time is shown as hours: minutes relative to mitotic entry. Scale bar, 20 μm. Representative of $n = 2$ experimental replicates. **c** Ratio of mitotic outcomes as shown in (**b**) after indicated duration of mitotic arrest (N cells from $n = 2$

experimental replicates). **d** Number of mitotic cell death and mitotic slippage from (**c**) in the indicated conditions within 2–6 h of mitotic arrest ($n = 2$ experimental replicates, unpaired two-tailed Fisher's Exact test). **e** Model of MAD telomere deprotection. (1) AURKB phosphorylates TRF1 at S354 and T358; (2) Survivin interacts with pS354 and pT358 to promote (3) AURKB phosphorylation of TRF2 at S62 and S65, enabling (4) BTR to dissolve t-loops. Source data are provided as a Source Data file.

We predicted AURKB-modification of the TRF2 basic domain enabled BTR-dependent dissolution of t-loop junctions, thereby linearizing chromosome ends to promote telomere DDR activation. In agreement, BLM depletion suppressed MAD-TIF in TRF2 shRNA IMR90 E6E7 hTERT cultures that also expressed TRF2-2D (Fig. 6f). BTR-dependent dissolution disentangles dHJs without nucleolytic cleavage. Should t-loop junctions assume a dHJ configuration, we anticipate BTR-dependent dissolution will linearize the telomeres without the rapid telomere deletions that occur when t-loop junctions are resolved through nucleolytic pathways[32,33]. In support, we observed no reduction in the telomere lengths of DDR-positive telomeres that arise through MAD-telomere deprotection when compared to their protected sister chromatid (Supplementary Fig. 6c, d). AURKB therefore phosphorylates TRF2-S65, and potentially S62, during mitotic arrest to compromise telomere protection against BTR.

To explore interaction dynamics between Shelterin, BLM, and AURKB, we benzonase-treated extracts from mitotically arrested HT1080 6TG cells to degrade the genomic DNA prior to co-IP (Supplementary Fig. 6e). BLM and AURKB recovery with Flag-TRF1, and AURKB recovery with Myc-TRF2, were Benzonase insensitive, consistent with direct interaction. BLM recovery with Myc-TRF2, however, was partially Benzonase sensitive, suggesting the BLM-TRF2 interaction requires co-localization on a common DNA substrate. Collectively the data are consistent with direct TRF1-BLM interactions facilitating BTR-dependent t-loop junction dissolution and MAD-TIF, without

telomere shortening, following TRF2 basic domain phosphorylation by AURKB.

## TRF2 modification impacts MAD lethality

A minority of cells experiencing mitotic arrest perish because of MAD telomere deprotection[22,24]. We previously demonstrated that enhancing MAD-TIF increased mitotic death occurrence and expedited time to death during mitotic arrest[22]. To test how TRF2 phosphorylation status impacts mitotic arrest outcomes, TRF2 shRNA IMR90 E6E7 hTERT cells expressing TRF2-WT, TRF2-S65A, or TRF2-S65D (Fig. 4d, e) were exposed to 100 ng mL$^{-1}$ colcemid and the outcomes analyzed through live imaging (Fig. 7a, b). As previously reported[22], TRF2 depletion significantly increased mitotic death in cells arrested in mitosis for 2 to 6 h (Fig. 7c, d). Suppressing MAD-TIF with TRF2-WT or TRF2-S65A significantly reduced cell death in the same range of mitotic arrest, whereas TRF2-S65D which promotes MAD-TIF did not (Fig. 7c, d). Similar non-significant trends were observed in the 6–10 h mitotic arrest window (Fig. 7c).

We also previously demonstrated that lethal replication stress in HT1080 6TG cells promotes mitotic arrest and death signaled in part through MAD-telomere deprotection[24]. We therefore depleted endogenous TRF2 in HT1080 6TG cultures and complemented with TRF2-WT or TRF2 carrying Ser65 mutations (Supplementary Fig. 7a). Cells were treated with 1 μM of the DNA polymerase inhibitor aphidicolin and cell outcomes measured through live imaging (Supplementary

Fig. 7b, c). Metaphase-TIF analysis was performed 40 h after aphidicolin addition (Supplementary Fig. 7b).

Exogenous TRF2 did not impact aphidicolin-dependent mitotic arrest (Supplementary Fig. 7d). As reported previously, TRF2-WT suppressed the enhanced metaphase-TIF in TRF2 shRNA cells treated with lethal replication stress (Supplementary Fig. 7e, f)[24]. Congruent with TRF2 phosphorylation promoting MAD-telomere deprotection, significantly more metaphase-TIF were present in TRF2-S65D expressing cells as compared to TRF2-WT or TRF2-S65A (Supplementary Fig. 7f). We note metaphase-TIF in aphidicolin treated cells can result from both interphase telomere replication stress carried into mitosis and MAD-telomere deprotection. Consistent with MAD-telomere deprotection promoting mitotic death, mitotic lethality was attenuated in TRF2-S65A expressing cells but not TRF2-S65D (Supplementary Fig. 7g). Further, the time of mitotic arrest to death was elongated when MAD-telomere deprotection was suppressed by TRF2-S65A, but was not affected by TRF2-S65D (Supplementary Fig. 7h). Together the data are consistent with AURKB-dependent TRF2 phosphorylation promoting mitotic death events signaled through MAD telomere deprotection.

## Discussion

Here we describe a mechanism of MAD telomere deprotection regulated by the CPC, shelterin, and BTR. Our data reveals that the CPC component AURKB phosphorylates TRF1 at Thr358 and potentially Ser354 during mitotic arrest (Fig. 7e [1]). This promotes CPC binding to phosphorylated TRF1 via Survivin (Fig. 7e [2]). During mitotic arrest, AURKB also phosphorylates the TRF2 basic domain on Ser65 and potentially Ser62 in a TRF1-dependent manner (Fig. 7e [3]). AURKB-dependent attenuation of the TRF2 basic domain enables BTR dissolution of t-loop junctions, resulting in linear telomeres and a localized ATM-dependent DDR (Fig. 7e [4]), which contributes to mitotic death. Through the identification of TRF1-dependent ATM regulation at somatic telomeres, and the participation of TRF1 and TRF2 in an active process of telomere deprotection, we reveal an unappreciated complexity in shelterin function.

### MAD-telomere deprotection kinetics

MAD-telomere deprotection requires four or more hours of mitotic arrest[25] and AURKB-dependent modification of multiple shelterin factors. During normal cell division, the CPC is subject to tight spatiotemporal regulation[35]. Mitotic arrest occurs during prometaphase or metaphase when the CPC is localized at kinetochores. We were unable to identify where shelterin and the CPC first interacted. However, observation of multiple kinetochore factors in our APEX2-Flag-TRF1 interactomics dataset suggests possible TRF1 localization at kinetochores during mitotic arrest. In support, TRF1 was previously implicated in centromere function[54]. Further, respective mitotic TRF1 phosphorylation on Thr371 in humans and Ser404 in mice by CDK1 and AURKB were proposed to dissociate TRF1 from the telomere substrate[55,56]. This may increase the pool of diffusive TRF1 in mitotic cells, facilitating stochastic and transient kinetochore interactions.

It is tempting to speculate that diffusive TRF1 stochastically interacts with kinetochores leading to CPC-directed phosphorylation of TRF1-S354 and/or T358. Phosphorylated TRF1, alone or in complex with the CPC via interaction through Survivin, may continue diffusive movement within mitotic cells. Eventually, the CPC-TRF1 complexes interact with telomeres and promote further AURKB-dependent shelterin phosphorylation. In agreement, we found TRF1 promotes TRF2-S65 phosphorylation in cultured cells, and that AURKB phosphorylates TRF1-T358 with faster kinetics than TRF2-S65 in vitro. This suggests TRF2 phosphorylation is the downstream and rate-limiting step.

We interpret the data to indicate that MAD-TIF are potentiated through CPC recruitment to shelterin via TRF1, which enables subsequent TRF2 phosphorylation. Directly attenuating basic domain protection via TRF2-S62 mutation circumvents the need for upstream TRF1 recruitment and accelerates MAD-TIF kinetics. This also indirectly supports AURKB modification of its TRF2-S62 consensus sequence being requisite for MAD-telomere deprotection. Because TRF2-S65D fails to accelerate MAD-TIF kinetics, this argues that additional TRF1-directed and AURKB-dependent TRF2-S62 phosphorylation is required for MAD-TIF to accrue. Additionally, because the CPC and BLM binding motifs on TRF1 overlap, TRF1 may also support BTR recruitment to mitotic telomeres, although experimental evidence is limited. We anticipate the slow kinetics of a sequential TRF1, CPC, TRF2, BTR pathway results in continued telomere protection during normal mitoses.

### The TRF2 basic domain regulates mitotic telomere protection

Data from several publications support protection of t-loop junctions by the TRF2 basic domain. Biochemical evidence demonstrates that the TRF2 basic domain binds 3- and 4-way DNA junctions[31,32,34], and also remodels oligonucleotides containing telomeric DNA into 4-way HJs[34]. Expression of a TRF2 mutant allele lacking the basic domain (TRF2ΔB) promotes rapid telomere deletions and free telomere DNA circles dependent upon homologous recombination (HR)[33]. Swapping the TRF2 basic domain with prokaryotic factors that bind 4-way DNA junctions restored TRF2 protection against such rapid telomere deletions[32]. Collectively the data indicate that when the TRF2 basic domain is absent, t-loop junctions are exposed to nucleolytic cleavage —a process termed t-loop HR[33]. T-loop HR is more pronounced in TRF2ΔB cells with concurrent BLM knockout, and unchecked t-loop HR can shorten telomeres sufficiently to promote chromosome fusions and genome instability[32].

Expression of exogenous TRF2ΔB, however, does not induce a strong interphase-TIF phenotype[33]. In agreement, all TRF2 basic-domain mutants examined here retained interphase protection. This argues that the physiological role of the basic domain is mitosis-specific protection. Understanding HR is informative as to why mitosis-specific t-loop junction protection is required. During HR, strand invasion can be followed by second-end DNA capture and dHJ formation[57]. HJs are resolved in mammalian cells through nucleolytic cleavage by the SLX4-SLX1-MUS81-EME1 complex[58] or GEN1[59]; enzymes whose HJ resolution activity is spatiotemporally restricted to mitosis[60]. Alternatively, dHJs can be dissolved without cleavage through the BTR complex[36] and BLM hyper-phosphorylation at the G2/M transition is required for full dissolution activity[61]. We anticipate that t-loop HR in mammalian interphase nuclei is tempered by inhibition or exclusion of HJ resolvase activity. When TRF2ΔB cells enter mitosis, however, the exposed t-loop junctions are likely subject to nucleolytic resolution and/or BTR dissolution.

We did not observe quantitative telomere shortening associated with MAD-telomere deprotection. We also did not observe unequal sister telomere intensity in mitotic chromosome spreads, the traditional qualitative evidence of rapid telomere deletions[33,62]. T-loop HR therefore remains suppressed following AURKB modification of the TRF2 basic domain. We interpret our findings to indicate that the basic domain protects t-loop junctions from both nucleolytic cleavage resulting in t-loop HR and BTR-dependent dissolution. TRF2 modifications during mitotic arrest derepress protection against BTR while maintaining protection against nucleolytic resolution. This enables t-loop junction dissolution, chromosome end linearity, and DDR activation without the risk of rapid telomere deletions.

### The molecular identity of t-loop junctions

The involvement of BTR in MAD-TIF formation is consistent with t-loop junctions assuming a dHJ configuration during mitotic arrest. It remains unclear, however, if dHJs are ubiquitous at the t-loop insertion point, or if these double four-way structures are only formed during

mitotic arrest. We currently favor the hypothesis that t-loop junctions typically assume a 3-way D-loop or 4-way single HJ that is bound and stabilized by the TRF2 basic domain. AURKB modification likely disrupts electrostatic interactions between the basic domain and its structured DNA substrate, allowing for branch migration, dHJ formation, and dissolution via BTR. The temporal dissolution of mitotic t-loops described here is reminiscent of RTEL1-dependent t-loop unwinding during S-phase[39]. Both pathways are restricted to a specific cell cycle window, executed by a helicase, and regulated through TRF2 phosphorylation. Notably, RTEL1 suppresses cross-over repair during HR through a proposed anti-recombinase mechanism that limits second-end capture and dHJ formation[63]. Inhibiting telomeric RTEL1 localization in S-phase promotes t-loop HR[39,64]. This is consistent with negative outcomes occurring when dHJs are present at the t-loop insertion point. Future cryo-EM studies of shelterin bound to structured DNA will ultimately clarify t-loop junction identity.

### Shelterin-mediated telomere deprotection

Shelterin is typically conceptualized as a chromosome-end protective complex where individual components inhibit specific DDR and/or double-strand break repair pathways. TRF2 is attributed as the sole factor in somatic cells responsible for t-loop formation and ATM suppression[15,16,30]. To our surprise, we found TRF1 is required for MAD-telomere deprotection. This revealed that shelterin is necessary for active telomere deprotection during mitotic arrest, and implicated TRF1 in somatic ATM and t-loop regulation under specific physiological conditions. The wider implication of our findings is that shelterin protective activities are more entwined than previously understood. Crosstalk between shelterin subunits may influence DDR outcomes previously considered under the control of a single telomere factor. Understanding crosstalk between shelterin components may inform future studies on aging-dependent telomere deprotection.

Finally, our study reveals shelterin is a dynamic complex that mediates both telomere protection and deprotection. Because NHEJ is repressed in mitosis[65], and TRF2 remains bound to deprotected mitotic telomeres to directly inhibit end-joining upon mitotic exit[29], there is no risk of telomere fusions from MAD-telomere deprotection[25]. MAD-telomere deprotection is therefore a highly coordinated mechanism that facilitates chromosome end linearity and telomere DDR activation without compromising genome stability through telomere shortening or fusion. Because mitotic arrest is a common response to genomic damage, including double-strand break induction[66] and replication stress[23,24], we anticipate MAD-telomere deprotection evolved as a failsafe to promote mitotic death or p53-dependent G1 arrest in genomically damaged cells.

## Methods
### Cell culture and treatment
IMR90 fibroblasts (female) were purchased from Coriell Cell Repositories, and HCT116 (male) and HT1080 (male) cells from ATCC (American Type Culture Collection). HeLa (female) cells were provided by Megan Chircop (CMRI), HT1080 6TG (male) cells by Eric Stanbridge (University of California, Irvine), and *TRF2^Floxed/Floxed Rosa26-CreERT2 pBabeSV40LT* MEFs (sex not determined) by Eros Lazzerini Denchi (National Cancer Institute)[13]. IMR90 E6E7 hTERT fibroblasts were produced by retroviral transduction of IMR90 with pLXSN3-16E6E7 and pWZL-hTERT[67]. IMR90 E6E7 hTERT fibroblasts, HCT116 cells, and HT1080 cells were grown in Dulbecco's Modified Eagle Medium (DMEM) supplemented with 10% fetal bovine serum (S1810-500, biowest), 200 mM L-glutamine, 7.5% NaHCO₃, 100 U mL⁻¹ penicillin, streptomycin, and 5 μg mL⁻¹ Plasmocin (InvivoGen) and maintained at 37 °C in 5% CO₂ and 3% O₂ for IMR90 E6E7 hTERT or in 5% CO₂ for HCT116 and HT1080. HT1080 6TG, HeLa-APEX2, HeLa-APEX2-TRF1, and MEF cultures were grown in Dulbecco's Modified Eagle Medium (DMEM) supplemented with 10% fetal calf serum (F9423, Sigma),

GlutaMax, and MEM non-essential amino acid solution (Gibco) and maintained at 37 °C in 7.5% CO₂ and 3% O₂. Cell Bank Australia verified cell line identity using short-tandem-repeat profiling, and cells were identified to be mycoplasma negative (MycoAlert, LT07-118, Lonza).

HT1080-6TG, HeLa-APEX2, HeLa-APEX2-TRF1, and MEF cells were synchronized in mitosis by treating cultures at 70% confluence with 2 mM Thymidine (Sigma) for 24 h. Thymidine was washed out with warm PBS, replacing with fresh media containing 150 ng mL⁻¹ Nocodazole (M1404, Sigma) for 14–16 h to arrest cells in mitosis. Rounded mitotic cells were detached from culture dishes and interphase cells by mitotic shake-off. Where indicated, cells were treated with 100 ng mL⁻¹ Colcemid (15212012, Gibco), 150 ng mL⁻¹ Nocodazole (M1404, Sigma), 40 nM Hesperadin (S1529, Selleck Chemicals), 10 μM XL413 (24906, Cayman Chemical), or 25 nM BAY181609 (A19868, Cayman Chemical).

### Plasmid cloning
Plasmids used in this study are listed in Supplementary Data 1. Complementary DNA encoding human TRF1 (NP_059523.2), the functional short isoform of TRF2 (XP_005256180.1, missing the first 42 a.a.)[68], and BLM (NP_000048.1) were generously provided by Jan Karlseder. The nuclear short isoform of TOP3A (Q13472-2) was artificially synthesized (Integrated NDA Technologies). All cDNAs were in-frame cloned downstream of blasticidin S-resistance gene and self-cleaving p2a sequences in a 3rd generation lentiviral plasmid vector. Flag and myc tags were added during PCR. pLPC-Myc and pLPC-Myc-TRF2 expressing the short TRF2 isoform were acquired from Addgene (Plasmids #12540 and #16066 respectively, kind gift of Tita de Lange). Mutations and deletions were introduced by site-directed PCR mutagenesis (TRF1, TRF2, BLM) or during artificial DNA synthesis (TOP3A) and cloned by the recombination-based HiFi assembly strategy (NEB). mScarlet and mClover/mClover3 were in-frame cloned at 5'-terminus of 3FL-TRF1 or TRF2 genes with (GGGGS)x2 linker sequence. Amino acid positions in each protein correspond to the indicated reference sequences.

APEX2 and APEX2-TRF1 vectors were created by assembling fragments into the pRRL-sin-cPPT-hPGK-eGFP-WPRE backbone (a kind gift from Leszek Lisowski). All constructs were cloned using PCR amplification of complimentary fragments and In-Fusion HD assembly (Takara). pRRL-hPGK-mCherry-P2A-PuroR-T2A-FLAG-APEX2 was created by PCR amplifying hPGK-mCherry-P2A-PuroR-T2A from a synthesised gene fragment purchased from GeneWiz and APEX2 from pcDNA3-FLAG-APEX2-NES (a gift from Alice Ting[69], Addgene plasmid # 49386) followed by insertion into PCR linearised pRRL backbone. pRRL-hPGK-mCherry-P2A-PuroR-T2A-FLAG-APEX2-TRF1 was created following the same approach, by fusing the same fragments with TRF1 PCR amplified from pLPC-myc-His-BirA-human TRF1 (a kind gift from Roderick O'Sullivan[70]).

Short hairpin RNA against *TRF1*, *TRF2*, *BLM*, *TOP3A*, *RMI1*, *RMI2*, *FANCJ*, *INCENP*, and *Survivin/BIRC5* were cloned into pLKO.1 vector by conventional restriction enzyme cloning of annealed oligonucleotides. The shRNA target sequences were as follows: Control 5'-CCTAAGGTT AAGTCGCCCTCGCTC-3', BLM 5'-TGCCAATGACCAGGCGATC-3'[71], *TRF1* 5'-CCCAGCAACAAGACCTTAATA-3'[72], *TRF2* 5'-GCGCATGACAATAAG CAGATT-3'[29], *TOP3A* 5'-GCTTCTCGAAAGTTGAGAATA-3'[73], *RMI1* 5'-CG ATCGAAGTATAGAGAGATT-3' (TRCN0000158474), *RMI2* 5'-CCATGA AAGTATGTGGGAACT-3' (TRCN0000143418), *FANCJ* 5'-CGTCAGAACT TGGTGTTACAT-3' (TRCN0000049914), *INCENP* 5'-AACTGTGACAGAT GATGCG-3'[74], and *Survivin/BIRC5* 5'-CCGCATCTCTACATTCAAGAA-3'[75]. shRNA-resistant silent mutations were as follows: *BLM* 5'-c GCt AAc GAt CAa GCc ATt-3', *TRF1* 5'-Ct CAa CAg CAg GAt tTg AAc A-3', *TOP3A* 5'-G Cc agc aGg AAa cTc cGg ATc-3', where lowercase denotes silent mutations. Exogenous TRF2 alleles did not carry shRNA-resistant silent mutations as exogenous TRF2 expression in the presence of TRF2 shRNA produces sufficient protein to rescue wild-type TRF2 function[29]. All plasmid sequences were confirmed by Sanger sequencing and are available upon reasonable request.

## Site-directed mutagenesis

Site-directed mutagenesis of plasmids listed in Supplementary Data 1 was carried out using KAPA HiFi HotStart premix (Roche) or Tks Gflex DNA polymerase (Takara Bio) by gradient touch-down PCR with cycler settings optimised for long-range synthesis. Mutagenesis primers are listed in Supplementary Data 2 and manufactured by IDT or Eurofins genomics. Following successful amplification, and confirmation by agarose gel electrophoresis, mutagenised amplicons were digested with DpnI (New England Biolabs (NEB)) to remove template DNA, phosphorylated by T4 polynucleotide kinase (NEB), ligated with Blunt/TA master mix (NEB) or DNA ligation kit (Takara) and transformed into Stellar™ competent cells (Takara Bio) or Mix & Go Stbl3 competent cells (Zymo Research). Following standard plasmid DNA preparation, mutagenesis was confirmed by Sanger sequencing.

## Viral packaging and transduction

Lentivirus particles for transduction of IMR90 E6E7 hTERT, HCT116, and HT1080 cells were produced by transfection of an expression plasmid with packaging and envelope plasmids gifted from Didier Trono (Addgene plasmid #12260) and Robert A. Weinberg (Addgene plasmid #8454), respectively, into HEK293FT or its derivative Lenti-X 293 T cells (632180, Takara). Media was replaced after 24 h and the viral supernatant was collected through filtration (0.45 μm pore, 25 mm, technolabsc inc.) at 48 and 72 h post-transfection and stored at −80 °C until transduction. The viral supernatant was complemented with 8 μg mL$^{-1}$ polybrene for transduction. Selection was carried out by using 10 μg mL$^{-1}$ Blasticidin-S (Funakoshi) for at least 5 days to obtain cells stably expressing target proteins. For knockdown experiments, cells were transduced with lentivirus carrying shRNA target sequences for two days and selected with 1 μg mL$^{-1}$ Puromycin (ChemCruz) for more than 3 days before experimental procedures.

For transduction of HeLa or HT080 6TG cells with APEX2, APEX2-TRF1, or shRNA vectors, high titre, purified pRRL/pLenti-derived lentiviral vectors were created by the CMRI Vector and Genome Engineering Facility. HeLa cultures were transduced with concentrated lentivirus at an MOI of 10 for 48 h in 4 μg mL$^{-1}$ polybrene (Sigma), then selected in normal growth media supplemented with appropriate antibody selection; APEX2 vectors 0.25 μg mL$^{-1}$ puromycin, shRNA vectors 1 μg mL$^{-1}$ or 5 μg mL$^{-1}$ Blasticidin-S or 400 μg mL$^{-1}$ Hygromycin. Following expansion, APEX2 cells were sorted for mCherry expression by the Westmead Institute for Medical Research Cytometry Facility (Sydney, Australia). Positively transduced cells were expanded and frozen in FBS with 10% DMSO at low passage. For FLAG-TRF1, high titre lentiviral vectors created by the CMRI Vector and Genome Engineering Facility were added to HT1080 6TG cell cultures supplemented with 4 μg mL$^{-1}$ polybrene for 48 h, then selected in normal growth media supplemented with 1 μg mL$^{-1}$ puromycin for 72 h.

Retroviral particles were produced by transfecting pLPC plasmids into low passage Phoenix-AMPHO cells (ATCC) at 90% confluence using Lipofectamine3000 (ThermoFisher) as per the manufacturer's instructions. Viral supernatants were removed at 24 and 48 h post-transfection, filtered through 0.45 μm syringe filters, and added to target cells in media containing 4 μg mL$^{-1}$ polybrene. Cells were grown for 48 h following retroviral transduction before selecting with 1 μg mL$^{-1}$ of Puromycin (ChemCruz) for more than 72 hours before experimentation. All retro/lentiviral transduced cells were maintained for short-term culture in 50% concentration of selection antibiotic to retain transgene expression.

## APEX2-TRF1 proximity biotin-labelling

We added 500 μM Biotin-Phenol (Iris Biotech, LS-3500) in a suspension of culture media containing 150 ng mL$^{-1}$ Nocodazole ± 40 nM Hesperadin (mitotic cells only) for 30 min. Cells were resuspended in 5 mL of media and 1 mM hydrogen peroxide was added to the cells for 60 s to initiate APEX2 labelling. Reactions were quenched with

3 × 3 min washes of quench solution containing 20 mM Sodium Ascorbate (Sigma), 10 mM Trolox (Sigma), and 20 mM Sodium Azide (Sigma). The cells were washed with PBS and frozen at −80 °C as a dry pellet, prior to streptavidin pull-down. For attached G1/S cells, all reactions were performed as described in the culture dish; following quenching, cells were removed by trypsin and frozen as a dry pellet.

## Recovery of biotinylated proteins for mass spectrometry

Cell pellets were resuspended cold GdmCl lysis buffer containing 6 M Guanidinium Chloride (Sigma) and 100 mM Tris-HCl pH 8.5 (Sigma) containing 10 mM TCEP (Sigma) and 40 mM IAM (Sigma) and lysate was heated to 95 °C for 2 × 5 min followed by homogenization with a tip probe sonicator for 2x 20 s cycles. Samples were diluted 1:1 in MilliQ H$_2$O and precipitated O/N at −20 °C with 4 volumes of acetone. Precipitated proteins were pelleted by centrifugation at 525 × $g$ for 5 min washed with 80% acetone at −20 °C, re-pelleted, and air dried. Precipitates were resuspended in GdmCl Lysis buffer (without TCEP and IAM) and protein concentration was assayed with BCA kit (Thermo-Fisher) according to manufacturer's protocol. Two mg of total protein from each sample was diluted with MilliQ H$_2$O to 1 M GdmCl in eqi-volume. Washed streptavidin magnetic beads (Pierce) were added 1:2 (bead:protein (w:w)) to each sample and incubated at 4 °C overnight at 1,200 RPM on a thermomixer (ThermoFisher). The beads were washed 3 × 30 s in GdmCl lysis buffer at 1200 RPM on the thermomixer at room temperature, followed by 2×30 sec in MilliQ H$_2$O.

## TRF1 peptide synthesis and pull-down

N-terminus biotin labelled 21-25-mer TRF1 peptides were chemically synthesised by the CMRI peptide synthesis facility. Peptides were synthesised via solid phase peptide synthesis (SPPS) on a Syro II automated peptide synthesiser. Resin (50 mg, Chemmatrix rink amide, 0.36 mmol g$^{-1}$, 200–400 mesh) was washed (3 × 5 min in dimethylformamide (DMF)), swelled in DMF, deprotected with piperidine (20% in DMF, 2 × 1 min, 1 × 3 min, 1 × 10 min), washed again (3 × 5 min in DMF), and the first Fmoc protected amino acid coupled. Subsequent coupling steps were performed with standard Fmoc and side chain protected amino acids and chemistry, using 1,3-diisopropylcarbodiimide (3 eq) and Oxyma (3.3 eq) in DMF and heating where stable until reaching a phosphorylated residue or unstable amino acid. Fmoc-Ser(HPO3Bzl)-OH and Fmoc-Thr(HPO3Bzl)-OH were used to introduce the relevant phosphorylated amino acids and these couplings were performed at room temperature. Subsequent coupling steps were also performed at room temperature with double couples and extended coupling times. The final coupling of biotin to the N-terminus was performed manually under the same room temperature extended double coupling conditions. Cleavage and deprotection were performed using a standard cleavage cocktail of TFA:TIPS:Water (95:2.5:2.5). Peptides were precipitated in diethylether, redissolved in ACN:Water (30:70 with the addition of 0.1% FA to the final volume) and lyophilised. Peptides were then purified by reverse phase HPLC (Shimadzu Nexera, C16, A: Water, B: ACN, 5–95% B over 45 min, 5 mL min$^{-1}$, 214 nm) and the target mass validated by MS (LCMS-2020, ESI-MS, [M + 2H]2+ in the positive and [M-H]- in the negative). Purity was calculated by peak integration. Fractions were pooled and lyophilised affording the final product as a dry powder.

Lyophilised peptides were dissolved in 10 mM Tris-HCl pH 7.5 to a concentration of 1 mg mL$^{-1}$ and bound to 0.6 mg of streptavidin magnetic beads (Pierce) for 2 h at 4 °C. Mitotically arrested HT1080 6TG cells were lysed in immunoprecipitation buffer containing 20 mM HEPES pH 8 (Sigma), 150 mM KCl (Sigma), 0.5 mM EDTA (Sigma), 0.2% IGEPAL (Sigma), 0.5 mM DTT (Sigma) and 5% glycerol (Sigma), supplemented with PhosStop (Roche) and cOmplete Protease Inhibitor Cocktail (Roche). Cell lysate was mixed at 1:100 (v:v) with Benzonase (Merck) to digest precipitated chromatin. Lysate was precleared with fresh streptavidin magnetic beads and assayed by BCA (ThermoFisher)

according to manufacturer's instructions. A total of 0.5 mg of pre-cleared lysate was added to each peptide-streptavidin bead mix and shaken for 3 h at 4 °C on a thermomixer at 1200 RPM. Non-specific interactors were removed by 2x washes with cold lysis buffer, followed by 3x washes with cold 100 mM ammonium bicarbonate solution to remove residual detergents. Samples were then processed as per proteomic sample preparation.

## Proteomic sample preparation

For APEX2/APEX-TRF1 samples, biotinylated proteins bound to streptavidin beads were resuspended in 100 mM ammonium bicarbonate (Sigma) and digested with 2 µg Trypsin (Promega) overnight at 37 °C shaking at 1200 RPM on a thermomixer. The supernatant of each sample was collected and acidified by adding up to 6 % trifluoroacetic acid (TFA) (Pierce) and 50% MS-grade acetonitrile (ACN) (Sigma). Precipitated lipids were removed by centrifugation at $20,800 \times g$ for 20 min, and the ACN was removed by vacuum centrifugation (Gene-Vac). Remaining supernatant was added to a house-packed stage-tip with 2x layers of C18 filter (Empore 3 M) activated with ACN. Stage-tips were washed 2x with 0.1% TFA, eluted in 40% ACN/0.1% TFA into a thin-walled 96-well plate (ThermoFisher), and dried using vacuum centrifugation (GeneVac). Peptides were resuspended in MS buffer containing 2% ACN in 1% Formic Acid (Sigma).

For TRF1 peptide pulldown, proteins bound to streptavidin beads were digested in 50 mM ammonium bicarbonate and digested with 0.5 µg of trypsin overnight at 37 °C shaking at 1200 RPM on a thermomixer. Samples were resuspended in 2% TFA in isopropanol (IPA) (Sigma) and loaded on stage tips packed in-house with 2x layers of styrene-divinylbenzene (SDB-RPS) filter (AFFINISEP). Samples were washed with 1% TFA/90% IPA and eluted in 5% ammonium hydroxide (Millipore)/80% ACN into a thin-walled 96-well plate. Desalted peptides were dried using vacuum centrifugation and resuspended in MS buffer.

## Liquid chromatography-tandem mass spectrometry

Resuspended peptides were loaded into a Ultimate3000 UPLC with an autosampler maintained at 4 °C (Dionex), before loading onto a house-pulled 40 cm 75 µm I.D. fused silica column (Polymicro) containing 1.9 µm Reprosil AQ C18 particles (Dr. Maisch) in a 50 °C column oven (Sonation) on a nano-ESI source attached to a Q-Exactive Plus tandem mass spectrometer (ThermoFisher). APEX2/APEX2-TRF1 peptides were separated using the UPLC running a binary buffer system of A (0.1% Formic Acid): B (90% ACN / 0.1% Formic Acid), over a gradient of 5-30% Buffer B over 150 min. An MS1 scan was acquired from 300 to 1600 m/Z (35,00 resolution, 3e6 AGC, 20 ms IT) followed by a data dependent MS2 scan with HCD dissociation in the Orbitrap (17,500 resolution, 1e5 AGC, 25 ms IT, 20 loop count). Thermo RAW files were acquired in centroid mode. TRF1 peptide pulldown samples were separated using the UPLC running a binary buffer system of A (0.1% Formic Acid): B (90% ACN/0.1% Formic Acid), over a gradient of 5–30% Buffer B over 50 min, followed by 30-60% Buffer B over 20 min. An MS1 scan was acquired from 300-1600 m/Z (35,00 resolution, 3e6 AGC, 20 ms IT) followed by a data-dependent MS2 scan with HCD dissociation in the Orbitrap (17,500 resolution, 1e5 AGC, 25 ms IT, 20 loop count).

## Bioinformatic analysis

Thermo RAW files were processed with MaxQuant (v1.6.0.16) in standard settings, using a Human proteome database (Aug. 2018 release). Proteins were quantified using label-free quantification (LFQ) with additional identification enabled by match-between-run window set to 1.5 min. MaxQuant outputs were processed and analysed in Perseus (v1.6.10.43). Briefly, common contaminants, reverse database IDs and protein IDs were removed. Datasets were filtered for proteins identified in <5 (APEX2 samples) or <3 (peptide pulldown samples) replicates of at least one sample group. Missing values of remaining proteins

were imputed using the entire matrix (APEX2 samples were additionally batch-corrected). To identify significantly enriched proteins, LFQ values for APEX2-TRF1 or phosphorylated peptides were tested against the respective APEX2 or non-phosphorylated peptide control samples using student's t-test, filtering for p-value < 0.05. Gene ontology enrichment analysis was performed using Enrichr interactive webtool[76]. Initial comparisons of nocodazole and nocodazole + hesperadin datasets in the APEX2 samples revealed no significant differences in protein enrichment via streptavidin pulldown. To reduce the overall complexity of the analysis, only the nocodazole dataset has been analysed and presented.

## Metaphase-TIF, Interphase-TIF, and interphase streptavidin staining

For metaphase-TIF assays, mitotic chromosome spreads were obtained by cytospin as described previously[77]. Briefly, cells were collected by trypsinization and swelled in hypotonic buffer (0.2% KCl, 0.2% Tri-sodium citrate) for 10 min at room temperature. Cells were spread onto superfrost plus microscope slides (Epredia) at 700 g for 10 min using a Cytospin 4 (Thermo Scientific) or Cellspin 1 (Tharmac) cytocentrifuge. Samples were fixed in 2% paraformaldehyde (PFA)/1xPBS for 10 min. For interphase imaging, IMR90 E6E7 hTERT, HeLa, or MEFs grown on 13 mm glass coverslips were fixed in 2% PFA/1x PBS for 10 min. Samples were incubated in KCM buffer (120 mM KCl, 20 mM NaCl, 10 mM Tris-HCl pH 7.5, 0.1% Triton X-100) for 10 min, followed by blocking in RNase A/ABDIL buffer (20 mM Tris-HCl pH 7.5, 2% BSA, 0.2% fish gelatin, 150 mM NaCl, 0.1% Triton X-100) at 37 °C for 30 min. For Interphase- and metaphase-TIF assays, cells were incubated with primary γ-H2AX antibody at 37 °C for 1 h. After 3 × 5 min washes in PBST buffer (1x PBS, 0.1% Tween-20) samples were incubated with secondary antibody. For interphase- and metaphase-TIF assays this was Alexa-568 goat anti-mouse IgG, 1:10,000 dilution at 37 °C for 30 min. For streptavidin staining we used streptavidin conjugated Alexa-488 at 1:1,000 dilution at 37 °C for 60 min. Samples were washed in PBST for 5 min three times, then fixed in 2% PFA /1x PBS for 5 min. Samples were dehydrated sequentially in 70%, 95%, and 100% EtOH, then incubated with 0.3 ng mL⁻¹ FAM-conjugated C-rich telomere peptide nucleic acid (PNA) probe (F1001, Panagene) in PNA buffer (70% formamide, 0.25% Blocking Reagent (NEN), 10 mM Tris-HCl pH7.5, 5% MgCl$_2$ buffer (82 mM Na$_2$HPO$_4$, 9 mM citric acid, 25 mM MgCl$_2$)) at 80 °C for 8 min. Following overnight incubation at 37 °C, samples were washed in PNA Wash A buffer (70% formamide, 10 mM Tris-HCl pH7.5) for 15 min twice and PNA Wash B buffer (50 mM Tris-HCl pH7.5, 150 mM NaCl, 0.8% Tween-20) for 10 min three times (0.2 µg mL⁻¹ DAPI in the second wash) and mounted with Vectashield PLUS Antifade (H-2000, Vector Laboratories) or ProLong Gold (Life Technologies).

For IMR90 E6E7 hTERT fibroblasts, HCT116, and HT1080 cells, images of interphase nuclei and/or metaphase spreads were taken with a 100x objective lens (PlanApo/1.45-NA oil) on a BZ-X710 microscope (KEYENCE) and analyzed by automated counting with Hybrid Cell Count and Macro Cell Count within the BZ-X Analyzer software (KEYENCE). For MEF and HT1080 6TG metaphase-TIF experiments, images were captured using a ZEISS AxioImager Z.2 with a 63x, 1.4NA oil objective and appropriate filter cubes, using a CoolCube1 camera (Metasystems). Automated metaphase finding and image acquisition for these experiments were done using the MetaSystems imaging platform as described elsewhere[29]. For imaging of HeLa and MEF interphase cells, images were captured using Zen software and a ZEISS AxioImagerZ.2, with a 63x, 1.4NA oil objective, appropriate filter cubes, and an Axiocam506 monochromatic camera (ZEISS).

## Visualization of TRF1 and TRF2 variants on mitotic chromosomes

IMR90 E6E7 hTERT fibroblasts were transduced with lentivirus carrying mScarlet-tagged 3FL-TRF1 and TRF2 variants. After 5 days of

blasticidin selection, cells were transduced again with mClover3-3FL-TRF1 or mClover-myc-TRF2. Cells expressing mClover3-3FL-TRF1 and mClover-myc-TRF2 were selected with puromycin for 3 days and blasticidin for 5 days, respectively. Cells with moderate expression of mScarlet and mClover/mClover3 were isolated using the SH800S cell sorter (Sony) and recovered for a week in culture. Endogenous TRF1 and TRF2 were depleted by simultaneous lentiviral shRNA infection followed by 3 days of recovery in the presence of puromycin. Cells were treated with 100 ng mL$^{-1}$ colcemid for 24 h and spread onto superfrost plus microscope slides (Epredia) at 700 g for 10 min by Cytospin 4 (Thermo Scientific). The cells on slides were fixed with 2% paraformaldehyde for 10 min following a 30 s pre-extraction with a pre-extraction buffer (0.5% Triton X-100, 20 mM HEPES pH7.9, 50 mM NaCl, 3 mM MgCl2, and 300 mM sucrose). The fixed slides were incubated with 0.2 µg mL$^{-1}$ DAPI for 5 min, rinsed with deionized water, dehydrated through a graded ethanol series (70%, 95%, 100%), and mounted with NPG anti-fade solution (4% n-propyl gallate, 100 mM Tris-HCl pH8.5, 90% glycerol). The mounted slides were imaged with a 100x objective lens (PlanApo/1.45-NA oil) on the BZ-X710 microscope.

**Telomere fluorescence intensity measurements**

To compare telomere FISH signal between sister chromatids, identifiable sister chromatid pairs were chosen from metaphase-TIF images, and telomere signals were manually quantified using the oval selection tool within ImageJ2 software (2.14.0/1.54 h). For each sister telomere pair, an ROI of minimal size was established, and the signals at the telomeres and in the background region between sister telomeres were measured as an average signal within the ROI. The background signal was deducted from the telomere signals before calculating signal ratio within each sister telomere pair and plotting as the log$_2$ value.

**Telomere-loop fluorescence imaging**

Samples were prepared for super-resolution imaging of telomere macromolecular structure as described elsewhere[11] with slight modification. Briefly, mitotic cells were pelleted at 1000 g for 5 min at 4 °C and washed with ice-cold nuclei wash buffer (10 mM Tris-HCl pH 7.4, 15 mM NaCl, 60 mM KCl, 5 mM EDTA, 300 mM sucrose). The sample was re-suspended in nuclei wash buffer in a 6-well non-tissue culture treated plate and incubated for 5 min while stirring on ice, in the dark, in the presence of 100 mg mL$^{-1}$ Trioxsalen (Sigma) diluted in DMSO. The material was then exposed to 365 nm UV light, 2–3 cm from the light source (model UVL-56, UVP), for 30 min, while incubated on ice with continuous stirring. After cross-linking, the material was centrifuged at 1,000 g for 5 min, washed with ice-cold nuclei wash buffer. Pre-warmed 37 °C spreading buffer (10 mM Tris-HCl pH 7.4, 10 mM EDTA, 0.05% SDS, 1 M NaCl) was added to the sample and deposited on an 18 × 18 mm 170 µm thick coverslip using a Cellspin1 (Tharmac) at 200 g for 1 min. Coverslips were fixed in −20 °C methanol for 10 min followed by 1 min in −20 °C acetone. Coverslips were rinsed with PBS then dehydrated through a 70%, 95%, and 100% ethanol series. Ethanol dehydrated coverslips were denatured for 10 min at 80 °C in the presence of C-rich telomere PNA probe conjugated to Alexa fluor 488 (Alexa488- OO-ccctaaccctaaccctaa, Panagene, F1004). PNA probe was prepared and diluted to 0.3 ng mL$^{-1}$ as described previously[41]. Following hybridization overnight in a dark humidified box, coverslips were washed twice in PNA Wash A (70% Formamide; 10 mM Tris-HCL pH 7.5) and thrice in PNA Wash B (50 mM Tris-HCL pH 7.5, 150 mM NaCl, 0.8% Tween-20) with gentle shaking. DAPI was added in the second wash of PNA wash B to stain the chromatin. Coverslips were rinsed in MilliQ water and dehydrated through a 70%, 95%, and 100% ethanol series. Slides were mounted in Prolong Gold (Life Technologies) in the presence of DAPI. Airyscan imaging was performed on a ZEISS LSM880 AxioObserver confocal fluorescent microscope fitted with an Airyscan detector and a Plan-Apochromat 63x 1.4 NA M27 oil objective. Alexa Fluor 488 labeled telomeres were captured with 4%

excitation power of 488 nm laser, 131 binning, detector gain of 950, and digital gain of 1 in super-resolution mode. A total of 5 z stacks (200 nm) were captured with frame scanning mode, unidirectional scanning, and line averaging of 2 in 1024 ×1024 pixels at 89.88 mm × 89.88 mm to scale. Z stacks were Airyscan processed using batch mode in Zen Black software (ZEISS). Images were maximum intensity projected and scored by eye with the researcher blinded to experimental conditions.

**Live cell imaging and analysis**

Cell Observer inverted wide field microscope (Zeiss) or BX-X710 fluorescence microscope (KEYENCE) with 20×/0.8 NA air objective was utilized to perform differential interference contrast imaging to visualise mitotic duration and outcomes. Cells were cultured on a glass bottom 24-well plate (MatTek Corporation) at 37 °C, 10% CO$_2$, and atmospheric oxygen provided by Zeiss Incubation System Module CELLS or stage-top chamber and temperature controller featuring a built-in CO$_2$ gas mixer INUG2-KIW (Tokai Hit). Cells were on the microscope for 3 h to achieve stability in the focal plane and images were captured every 6 to 15 min up to 63 h using an Axiocam 506 monochromatic camera (Zeiss) and Zen software (Zeiss) or BZ-H3XT time-lapse module (KEYENCE). For all movies, mitotic duration was scored by eye and calculated from nuclear envelope breakdown until cytokinesis, multipolar division, slippage, or mitotic death. The imaging analysis was performed using Zen or Fiji software.

**Immunoprecipitation**

Trypsinised HT1080-6TG cells were resuspended in cold non-denaturing lysis buffer containing 20 mM Tris-HCl pH 8.0, 137 mM NaCl, 1% IGEPAL, 2 mM EDTA supplemented with PhosStop and cOmplete Protease Inhibitor cocktail. Nuclei were mechanically dissociated through a 27 G syringe before removing insoluble chromatin by centrifugation at 20,800 × g. Protein concentration was determined by BCA per the manufacturer's instructions and precleared with 50 µg of protein A/G magnetic beads (Peirce) for 30 min at 4 °C mixing at 1200 RPM on a thermomixer. Washed protein A/G beads were incubated with primary antibody in non-denaturing lysis buffer containing 100 µg mL$^{-1}$ BSA for 1 hour at 4 °C on a thermomixer at 1200 RPM. Antibody bound beads (0.1–0.2 mg) were added to total protein lysate (0.5–2 mg) and shaken for 2 hours at 4 °C mixing at 1200 RPM on a thermomixer (for anti-TRF1 pulldowns, bead incubation was increased to overnight, on a nutator). Non-specific interactors were removed by 3 × 1 min washes with IP wash buffer containing 20 mM Tris-HCl pH 8.0, 300 mM NaCl, 1% IGEPAL, and 2 mM EDTA. Proteins were eluted by mixing in 2xLDS buffer (-EDTA) containing β-mercaptoethanol at RT for 5 min. Immunoprecipitations were analysed by western blotting, comparing to 0.5–1% (w:w) of input lysate. Where indicated, extracts were treated with 50 U mL$^{-1}$ Benzonase (Millipore) on ice after preclearing but prior to immunoprecipitation.

**In vitro kinase assay**

Thymidine synchronised HT1080 6TG cells was processed as per immunoprecipitation protocol using 80 µg of total lysate per kinase reaction, incubating overnight with antibody-bead mix. Following removal of non-specific interactors with IP wash buffer, beads were rinsed with kinase assay buffer containing 25 mM MOPS pH 7.2, 25 mM MgCl$_2$, 5 mM EGTA, 5 mM EDTA, and 0.25 mM DTT diluted 1:5 in a sterile 50 µg mL$^{-1}$ BSA solution. Beads were resuspended in diluted kinase assay buffer containing 0.5 mM ATP and recombinant Aurora B kinase (AbCam). Kinase reactions were shaken at 30 °C for 10 min at 1000 RPM on a thermomixer and quenched with 4x LDS buffer (-EDTA) containing β-mercaptoethanol. Assays were analysed and detected per western blotting protocol. Resultant immunoblots were analysed by densitometry corrected for background and loading against total TRF1 or 2. Enzyme kinetics were estimated using

non-linear Michaelis-Menten fitting in Prism, reporting $K_m$ within a 95% confidence interval.

## Western blotting

For IMR90 E6E7 hTERT fibroblasts, whole-cell extracts were prepared with lysis buffer (20 mM Tris-HCl pH8.0, 2.5 mM MgCl$_2$, 150 mM KCl, 0.5% NP40, 1 mM DTT, 0.2 mM PMSF, 1x protease inhibitor (ROCHE), and 1x phosphatase inhibitor (ROCHE)). Following centrifugation for 10 min at 13,000 g and supernatant collection, the total protein concentration was measured by spectrometry using a BioRad Protein Assay Dye Reagent (5000006JA, BioRad). Approximately 50 μg of cell lysate was resolved using 4–20% Mini-PROTEAN TGX precast gels (BioRad) in Tris-Glycine running buffer for 80 minutes at 100 V. Proteins were transferred to 0.45 μm PVDF membranes (Millipore) previously activated with 10% methanol using a Trans-Blot Turbo Transfer System (BioRad) for 7 min. Membranes were blocked with Blocking Buffer (Nacalai) for 30 minutes at RT and incubated with primary antibodies overnight at 4 °C. After 3 × 5 min washes in 1x TNT buffer (10 mM Tris-HCl pH8.0, 150 mM NaCl, 0.05% Tween-20), the membrane was incubated with secondary antibodies diluted in 1x TNT buffer for 1 h at RT. Following 3 × 5 min washes with 1x TNT, signal detection was developed by Chemi-Lumi Super One (Nacalai) and imaged by LAS-3000 imager (Fujifilm).

For HeLa, HT1080 6TG, and MEFs whole cell extracts were prepared by lysing cells in 4xLDS buffer (-EDTA) containing β- mercaptoethanol and 500 U Benzonase (Millipore) at 10,000 cells μL$^{-1}$ for 10–15 min at RT, followed by denaturation at 65 °C for 10 min. Proteins were resolved using precast 1.5 mm 4–12% Bis-Tris NuPage gels (ThermoFisher) and transferred to 0.2 μm nitrocellulose membranes (GEHE10600008, Amersham). For total proteins, membranes were blocked in 5% skim milk powder/1x TBS containing 0.1% Tween-20 (TBS-T), for phospho-specific antibodies membranes were blocked in 5% BSA/TBS-T for 1 h at RT with gentle shaking, for total biotin blots membranes were blocked overnight in 3% BSA/TBS-T without primary antibodies. Membranes were incubated in primary antibodies diluted in 5% BSA/TBS-T overnight followed by incubating with secondary antibodies in 5% skim milk powder/TBS-T for 1 h at room temperature. Following block, total biotin blots were treated with streptavidin-HRP at a concentration 0.3 μg mL$^{-1}$ in 3% BSA/TBS-T for 1 h at RT. HRP signal was detected in 9:1 mix of Clarity:Clarity Max ECl reagent (BioRad) and captured using a Chemidoc Touch imager (BioRad).

## Antibodies and recombinant proteins

Antibodies and their dilution for Western blotting were as follows: TRF2 (NB110-57130SS, Novus Biologicals), 1:1000; TRF1 (sc-56807, SantaCruz), 1:1000; TRF1 rabbit polyclonal antibody (Ishikawa lab, used in Supplementary Fig. 2), 1:1,000; BLM (NB100-214, NovusBio), 1:5000; Flag (F1804, Sigma), 1:2000; Myc (9B11 Cell Signalling Technology), 1:1,000; TOP3A (14525-1-AP, Proteintech), 0.3 μg mL$^{-1}$; StreptAvidin-HRP (S911, ThermoFisher), 1:1000; INCENP (ab-12183, AbCam), 1:2,000; Aurora B (ab-2254, AbCam), 1:1,000; Borealin (ab74473, Abcam), 1:1000; Survivin (NB500-201, Novus Biologicals), 1:1000; beta-Actin (A5441, Sigma), 1:20,000; Actin (MAB1501R, Millipore), 1:10,000; Vinculin (V9131, Sigma), 1:5000; GAPDH (MAB374, Millipore), 1:5000; GAPDH (M171-3, MBL), 1:1000; Goat anti-mouse HRP (P044701, DAKO Agilent), 1:5,000; and Goat anti-Rabbit HRP (P044801, DAKO Agilent). For immunofluorescence we used: γ-H2AX (Clone 2F3, Biolegend) 1:200 or (05-636, Merck Millipore), 1:500; Alexa-568 goat anti-mouse (A11031, Invitrogen); 1:10,000; and Streptavidin Alexa-488 (S32354, Invitrogen), 1:1000.

Polyclonal rabbit antibodies against phopsho-TRF1 and TRF2 were generated by Genscript. Briefly, animals were immunized with peptides corresponding to pTRF1-T358 (SRRA(p)TESRIPVSKS), and pTRF2-Ser65 (ASRS(p)SGRARRGRHEC). Antibodies were purified by antigen affinity purification selecting for minimized cross adsorption and validated using indirect ELISA. For western blotting, antibodies were diluted in 5% BSA/TBS-T at a concentration of 0.2 μg mL$^{-1}$.

## Reverse transcription quantitative PCR (RT-qPCR)

RNA from control and knockdown cells (Supplementary Figs. 3b and 6a) was harvested on day 3 of puromycin selection following lentiviral transduction, using the RNAeasy Mini Kit (Qiagen). RNA concentrations were quantified by Nanophotometer N50 (Implen), and a total of 0.33 μg RNA was used for cDNA synthesis using Thermal cycler Veriti (Thermofisher). Quantitative PCR (qPCR) was performed in technical duplicates using either Thunderbird® Next SYBR™ qPCR Mix (Toyobo) (Supplementary Fig. 3b) or Power SYBR® Green Master Mix (Thermofisher) (Supplementary Fig. 6a), with gene-specific primers on a StepOnePlus™ PCR System (Thermofisher). Ct-values were normalized to *ACTB* or *GAPDH* and fold changes were calculated by the the the $2^{-\Delta\Delta CT}$ method. Primers used are as follows:

*ACTB*_F, 5′-CCAACCGCGAGAAGATGA-3′;
*ACTB*_R, 5′-CCAGAGGCGTACAGGGATAG-3′;
*GAPDH*_F, 5′-CCTGCACCACCAACTGCTTAG-3′;
*GAPDH*_R, 5′-GGTCATGAGTCCTTCCACGATAC-3′;
*FANCJ*_F, 5′-CAGGCCCTTGGTAGATGTATTAG-3′;
*FANCJ*_R, 5′-TGCTGCCGTACCCATTTAG-3′;
*BLM*_F, 5′-TGGTGCGGAAGTGATTTCAGT-3′;
*BLM*_R, 5′-CTCCTCAGCGGCACTTCTTC-3′;
*TOP3A*_F, 5′-TGAGGATGATCTTTCCTGTCG-3′;
*TOP3A*_R, 5′-GCCTGGCCATACAGATGATAA-3′;
*RMI1*_F, 5′-GCGGTTCCTGTCCTTACAGT-3′;
*RMI1*_R, 5′-CTATTACCACGAGGAACAGCAG-3′;
*RMI2*_F, 5′-ATGCAGGGCAGGGTAGTG-3′;
*RMI2*_R, 5′-CCCACATACTTTCATGGATGG-3′;
*INCENP*_F, 5′-CAAGAAGACTGCCGAAGAGC-3′;
*INCENP*_R, 5′-TCAGGAGCCTCTCCAGGTAA-3′;
*Survivin*_F, 5′-CCCTGCCTGGCAGCCCTTTC-3′;
*Survivin*_R, 5′-CTGGCTCCCAGCCTTCCA-3′.

## Statistical analysis and figure preparation

GraphPad Prism 9 was used for all the statistical analysis in this study. Box plots are displayed using the Tukey method. Here the box extends from the 25th to the 75th percentile with the median represented by a line. The upper whisker is the 75th percentile + (1.5 × the inner quartile range) or the largest value if no data points are outside this range. The lower whisker is the 25th percentile − (1.5 × the inner quartile range) or the smallest data point if no data points are outside this range. Data outside of the whisker range are shown as individual points. P-values are indicated in each figure panel if applicable. We considered the *p*-value less than 0.05 as statistically significant. The number of experimental replicates is indicated in the figure legends. Comparisons between two groups were performed using Mann–Whitney test. Multiple comparisons were performed using Kruskal-Wallis followed by Dunn's test unless otherwise indicated in the figure legends. For mitotic fate analysis, pairwise comparisons were performed using Fisher's Exact Test, with significance thresholds adjusted using the Bonferroni correction (0.05/number of comparisons); only comparisons meeting this adjusted significance threshold (0.0125) were considered statistically significant. For metaphase-TIF analysis in IMR90 E6E7 hTERT fibroblasts, images were taken blindly by lab technicians and quantified using automated BZ-X Analyzer software. For metaphase-TIF analysis in all other cell lines, and for the analysis of telomere macromolecular structure, scientists were blinded to the experimental conditions. Figures were prepared using Adobe Illustrator and Affinity Designer software. The mass spectrometer in Fig. 3a and Supplementary Fig. 1c was created in BioRender (Cesare, T. (2025) https://BioRender.com/r86h834). All other drawings were created by the authors.

## Materials

Unique reagents generated here are available upon request. Please direct all enquires to the corresponding authors MTH (makoto.hayashi@ifom.eu) and AJC (tcesare@cmri.org.au).

## Reporting summary

Further information on research design is available in the Nature Portfolio Reporting Summary linked to this article.

## Data availability

The mass spectrometry outputs, and MaxQuant analyses in this paper, are available on the PRIDE database under the accession code PXD043281. The remaining datasets generated and/or analysed during the current study are available from M.T.H. and A.J.C. upon request. Source data are provided with this paper.

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

## Acknowledgements

We thank Will Hughes, Scott L. Page, and the CMRI ACRF Telomere Analysis Centre supported by the Australian Cancer Research Foundation for microscopy infrastructure and support; the Westmead Institute of Medical Research Flow Cytometry Centre, supported by the Cancer Council NSW and the Australian NHMRC for cell sorting; Mark Graham and George Craft and the CMRI Biomedical Proteomics Centre for Mass spectrometry infrastructure and support; the CMRI Peptide Synthesis Facility for their assistance; the Drug Discovery Centre supported by the iSAL (Innovative Support Alliance for Life Science), Kyoto University, for the cell sorter; Andrea Ruelas-Gonzalez and Yuya Nishida for experimental support with metaphase-TIF analysis; Yumi Hayashi for assistance with molecular cloning; Ylli Doksani for critical reading of the manuscript. Members of the Hayashi and Cesare labs are thanked for their suggestions and discussion. A.J.C. is supported by an Australian Research Council Future Fellowship (FT210100858). This project was supported by grants from the Australian NHMRC to A.J.C. and M.T.H. (1106241) and A.J.C. (1162886); the Australian Research Council to A.J.C. (DP210103885, DP240101869); an Australian Medical Research Future Fund award to A.J.C. (2007488); Japan Foundation for Applied Enzymology to M.T.H.; Grant-in-Aid for Young Scientists (A) (16H06176) to M.T.H.; Grant-in-Aid for Scientific Research (B) (20H03183) to M.T.H.; the Hakubi project grant to M.T.H.; the Takeda Science Foundation to M.T.H.; and institutional funding from CMRI (to N.L. and A.J.C.) and Kyoto University (to F.I. and M.T.H.).

## Author contributions

D.R.Z., S.R., M.T.H., and A.J.C. conceived the study. D.R.Z. and S.R. performed most of the experiments and data analysis in this manuscript. S.R. completed the mass spectrometry. R.R.J.L. completed the t-loop

imaging and analysis, and other experimentation. A.B.R. performed the peptide synthesis. M.T.H. and A.J.C. assisted with data analysis throughout. S.G.P, and B.J.E.L., created and verified plasmids and cell lines, and along with S.K. and L.F. assisted with minor experimentation throughout. N.L., F.I., M.T.H., and A.J.C. supervised junior members of the research team and secured funding. S.R., D.R.Z., M.T.H., and A.J.C. wrote the manuscript with editorial input from R.R.J.L., S.G.P., B.L., N.L.

## Competing interests

The authors declare no competing interests.
