## [Transparent Peer Review file · Nature Communications]

A CPC-shelterin-BTR axis regulates mitotic telomere deprotection

Corresponding Author: Professor Anthony Cesare

Version 0:

Reviewer comments:

Reviewer #1

(Remarks to the Author)

Romero-Zamora et al. present a novel aspect of MAD regulation at telomeres of cell arrested in mitosis. The phosphorylation of TRF1 and TRF2 by the CPC component AURKB appears to be important for the induction of MAD. BLM appears to be linked to the phosphorylation of TRF1 and TRF2 and the formation of linear telomeres. BTR activity drives MAD telomere deprotection.

The manuscript contains very relevant new insights into MAD regulation. Experiments and technological approaches are very appealing and straight forward. New functionally relevant sites in TRF1 and TRF2 have been identified. However, the data of the manuscript is difficult to follow and - to my opinion - some central elements that would allow the authors' conclusions are missing. This is probably also due to the authors' decision by the authors to integrate the role of both, P-TRF1 and P-TRF2 into the same manuscript. This generated a large amount of data that is not always easy follow and at some points do not appear to be interconnected. Given the elegant approach and scientific relevance I suggest an intensive work-over of the manuscript.

Please find below my suggestions to the authors of the manuscript:

It would be important to understand the localization of relevant P-TRF1 and P-TRF2 versions in cells (ChIP, Immunofluorescence or other approach). Are P-TRF1 or P-TRF2 versions enriched at telomeres in arrested mitotic cells? Do the P-TRF1 or P-TRF2 versions locate to other regions in the genome (for example the centromere regions)?

The interconnection of P-TRF1 related functions with P-TRF2 related functions in MAD are not clear to me. What are overlapping functions? What functions specific for TRF1 or TRF2? Do P-TRF1 and P-TRF2 communicate with each other during MAD?

Given the focus on CPC in the title of the manuscript authors should better demonstrate the relevance of CPC components for MAD. CPC loss of function experiments (RNAi) with TIF analysis of metaphase arrested cells could be an option. In addition, RNAi mediated loss of individual CPC components may reduce TRF1 and TRF2 phosphorylation at the relevant sites.

A possible phosphorylation (CPC) dependent interaction of BLM with TRF1 and/or TRF2 not clear to me. For example it was not tested whether TRF2 wt and mutant versions can immunoprecipitate BLM and/or CPC components.

Data from TRF1 IP experiments (Fig. 3D) show that BLM interacts in an efficient manner also with mutant TRF1 versions, independently on the amount of AURKB in the immunoprecipitate. I am not clear about the role of the TRF1 (TRF2) phosphorylation for BLM recruitment. In addition, the role of other CPC components (Borealin, Survivin, TOP3A) for BLM interaction (or BRT function) is not clear.

Very nice data was shown on telomere linearization in the part focusing on TRF2. However, the eventual impact of mutant TRF1 versions on telomere linearization remains unclear.

General comment on TIF data on metaphase chromosomes: The representative images are very small and termini of individual chromosome cannot be identified. Thus, it is not really evident that DNA damage occurs exclusively at telomeres. Given that CPC activity is central for the manuscript, authors should make clear that damage is limited to telomeres or also

occurring at other sites.

Other more specific questions related to individual figures of the manuscript:

Figure 1A: the APEX2-TRF1 is highly overexpressed in cells used for pull down experiments. Authors should comment eventual ectopic effects result in from TRF1 overexpression.

Fig.1F: Authors chose BLM as main candidate for regulating MAD activity by binding to TRF1. Fig. 1F contains other candidates of interest – for example FBXO18 appears to be interesting as well. Other classic telomere regulators such as WRN or ATRX show up in the candidate list. Are there other arguments for focusing on BLM or excluding other candidates?

Figure 2 and 3: Experiment on the generated phosphor-specific TRF1 antibody and the phosphor-peptide pull down are very convincing. Nevertheless, it would be important to get an idea on the % of total TRF1 and also TRF2 protein subjected to phosphorylation by AURKB.

Figure 2B and Supplementary figure 2D: What are the total levels of endogenous TRF1 and mutant TRF1 variants in the experimental cells? Suppl. Fig 2D shows only ectopic TRF1 levels. It would be important to know the proportion of wt TRF1 and mutant TRF1 in experimental cells (as shown for experiments with mutant TRF2 versions).

Figure 2B: Results of metaphase TIF abundance are very nice. Given the importance of this data the impact of most important mutant TRF1 version should be validated in a different cell model (HT1080 for example).

Fig. 2D: Western blotting experiment was performed in TRF1 overexpression conditions. Can phosphorylation also be found on endogenous TRF1?

Additional question on data related to Figure 2:

1. Do the used TRF1 variants (Figure 2) locate to telomeres or are they located at other regions of chromosomes (IF, ChIP)
2. TRF1 is a suppressor of replication stress and telomere fragility. Can authors show data on these features focusing on most relevant TRF1 mutants? Do these features potentially contribute to MAD? Is there DNA damage related to TRF1 mutants in interphase cells?
3. What is the final outcome of MAD impairment by most relevant TRF1 mutants on genome stability and cell survival? A possible readout is presented in Fig. 7 for the role of TRF2 on MAD.

Fig. 3D: TRF2 appears to bind with reduced efficacy to mutant TRF1 versions TRF1deltaHD and TRF1-2D. TRF2 binding to TRF1-2A appears to be normal. It would be interesting to validate this finding by ChIP on telomere repeat containing chromatin as this may have an impact on the formation of linear telomeres.

Figure 4F, Supplementary Figure 4E: Do the mutant TRF2 versions locate to telomeres or other locations along chromosomes?

Does the expression of TRF2 variant sequences have an impact of TRF1 localization interphase cells or metaphase cells?

Supplementary figure 4A. I do not understand why the pTRF2(S65) band in lane 10+11 runs higher than in other lanes (top blot).

Western blotting experiment was performed in TRF2 overexpression conditions. Can phosphorylation also be found on endogenous TRF2?

Supplementary Figure 4C:

- TRF2 levels shown by western blotting appear to be heterogeneous in the experimental cell lines. Can this impact on DNA damage data shown later in the manuscript?

Fig. 4F; Supplementary figure 4G: Given the importance of the demonstration of TRF2 phosphorylation on mitotic TIF formation, it is suggested to validate most relevant TRF2 variants in an independent cell model (HT1080).

Supplementary figure 4G: The effect of ectopically expressed, wt TRF2 is missing in the quantification of TIF data – that makes it difficult to get an idea on the effect of the tested TRF2 mutant proteins.

In addition, TIF data of all samples appears to be highly variable at the 24hrs time point. Numbers of metaphase should be increased to provide more convincing data.

Reviewer #2

(Remarks to the Author)

Comments on Zamora et al.; NCOMMS-24-09072

Mitotic arrest that is accompanied by a partial telomere deprotection has been reported since a while. Intriguingly, this partial deprotection involved resolution of the t-loops in a telomere length independent manner and did trigger an ATM-dependent DNA damage response. Yet, it does not yield telomere rapid deletion events or end-to-end fusions, which would be hallmarks of telomere deprotection. This latter classically was ascribed to the telomere binding protein TRF2 only. However, it remained unclear how the mitotic arrest dependent (MAD) phenomenon is triggered inside cells and what factors are associated with it.

Using APEX methodology combined with proteomics, here it is shown that this MAD partial deprotection involves two major complexes, first the chromosome passenger complex (CPC, with the AURKB kinase as executing enzyme). The AURKB was already known to be somehow involved and the authors then go on to assess the functional importance and the targets of AURKB, which remarkably yielded TRF1 and TRF2. Specific phosphorylation sites were mapped *in vitro* / *in vivo*, and P-specific antibodies allowed further refinement of the mechanistic impact of those phosphorylations. This in turn led to the discovery of, secondly, the BTR complex as being associated with MAD telomeres and most likely the agent that mediates the partial deprotection.

These results advance our understanding of telomere biology in several very important ways: the results allow the description of a pathway of TRF1-P that attracts the CPC and allows phosphorylation of TRF2 and a specific type of partial telomere deprotection (Fig. 7g). Secondly, the data postulate that both, TRF1 and TRF2 are involved in this deprotection pathway, which at the same time is a partial protection pathway (re no rapid deletions and end-fusions). The TRF1 protein in this context is required to bring the CPC with the AURKB to the telomeres and without that, TRF2 is not phosphorylated and does not elicit a MAD-TIFs. Therefore, the interplay between the shelterin proteins TRF1 and TRF2 is more intricate than anticipated and previously stipulated.

The experiments are extremely well designed and executed at an amazing level of technical quality and precision. This reviewer also appreciated the efforts to describe the sometimes quite involved setups in a clear and transparent way. To me, this manuscript clearly is breaking new ground on several fronts and deserves a very high visibility.

There are only a few minor comments / suggestions for corrections.

1) Fig 1D and S1G: line 150 ff in the text says that "...the CPC components AURKB, INCENP, and CDCA8 (Borealin), were strongly enriched within the Flag-APEX2-TRF1 samples". Upon inspection of the data, I could not validate the claim for AURKB: it seems to be in the non-specific domain in the volcano plot and barely distances itself from baseline in Fig. S1G. AURKB has been identified previously for this pathway (Ref 25) and it makes sense to follow that up. But the description here does not match the data.

2) While I agree that the mitotic death phenotype induced by MAD-TIF is a significant read-out for the consequences of this effect, I am not enthusiastic about the way the experiments were done in Fig. 7. TIF induction is by lethal dose of aphidicolin, which is a DNA polymerase inhibitor and well known for its effects on S-phase cells. Therefore, as the authors acknowledge in the text, the vast majority of TIFs scored in this experiment probably has nothing to do with MAD-cell death but is associated with S-phase catastrophe. Therefore, I am unsure whether the cell-death suppression phenotype in Fig. 7E is related to MAD-cell death or not. I would suggest performing this experiment in a genuine MAD background (microtubule inhibitor for example) such that we can be more confident of the MAD associated cell death.

Minor issues:

3) Please name TRF1 and TRF2 consistently: in volcano plot and in S1E they are labeled TERF1 and TERF2.

4) Abstract lines 20-32 are a very difficult read. Please break up sentence for clarity.

5) Perhaps mark the 11 helicases in Fig. 1F and mentioned in the text (line 166).

Other questions that I feel are pertinent but that do not need to be addressed with experiments:

- Why/how is TRF1-p by AURKB specific to M-phase?

- Could you do pull-back experiments to evaluate when the MAD arrest and cell death phenotype become irreversible? I.e. which is the execution step for the cell death.

Version 1:

Reviewer comments:

Reviewer #1

(Remarks to the Author)

The manuscript entitled “A CPC-shelterin-BTR axis regulates mitotic telomere deprotection“ describes a novel pathway that connects the CPC complex with TRF1 and TRF2 in controlling telomere de-protection during mitotic arrest. The results shown in the revised version of this manuscript are highly relevant for the field of telomere biology and genome stability and absolutely merit publication in Nature Communications.

The revised version of the manuscript contains a complete set of experiments that confirms the author’s claims. Methods and data analysis are state of the art and meet the expected standards of the field. The questions related with the first version of the manuscript were addressed with new experiments and/or discussed in sufficient detail.

For this reason, I strongly support publication of the final version of the manuscript.

Reviewer #2

(Remarks to the Author)

I have now read the revised version of this manuscript and I am very happy with all the responses and additions provided by the authors. This manuscript in my opinion is now ready to be published.

The images or other third party material in this Peer Review File are included in the article’s Creative Commons license, unless indicated otherwise in a credit line to the material. If material is not included in the article’s Creative Commons license and your intended use is not permitted by statutory regulation or exceeds the permitted use, you will need to obtain permission directly from the copyright holder.

Point by point response to the reviewers' comments.

We sincerely express our gratitude to the reviewers for their thoughtful assessment of our work. Their comments were well received, and the resulting changes have improved manuscript quality. Our responses to their comments/questions are in blue. The response to Reviewer 1 starts on Page 1, the response to reviewer 2 on page 12.

Author list - As a result of their contribution during revision, Shunya Kosaka and Lucy French were added to the author list. This change was agreed upon by all authors.

REVIEWER COMMENTS

Reviewer #1

Romero-Zamora et al. present a novel aspect of MAD regulation at telomeres of cell arrested in mitosis. The phosphorylation of TRF1 and TRF2 by the CPC component AURKB appears to be important for the induction of MAD. BLM appears to be linked to the phosphorylation of TRF1 and TRF2 and the formation of linear telomeres. BTR activity drives MAD telomere deprotection. The manuscript contains very relevant new insights into MAD regulation. Experiments and technological approaches are very appealing and straight forward. New functionally relevant sites in TRF1 and TRF2 have been identified. However, the data of the manuscript is difficult to follow and - to my opinion - some central elements that would allow the authors' conclusions are missing. This is probably also due to the authors' decision by the authors to integrate the role of both, P-TRF1 and P-TRF2 into the same manuscript. This generated a large amount of data that is not always easy follow and at some points do not appear to be interconnected. Given the elegant approach and scientific relevance I suggest an intensive work-over of the manuscript.

We thank the reviewer for their thoughtful and constructive feedback and appreciate their recognition of the novel insights and technical approaches presented. We have edited the manuscript and included new data as suggested which strengthen our conclusions and improve the clarity of the work.

Please find below my suggestions to the authors of the manuscript:

1. It would be important to understand the localization of relevant P-TRF1 and P-TRF2 versions in cells (ChIP, Immunofluorescence or other approach). Are P-TRF1 or P-TRF2 versions enriched at telomeres in arrested mitotic cells? Do the P-TRF1 or P-TRF2 versions locate to other regions in the genome (for example the centromere regions)?

We thank the reviewer for this helpful suggestion. Unfortunately, the TRF1-pT358 and TRF2-pS65 antibodies are incompatible with ChIP experiments.

As an alternative approach, we tagged exogenous TRF1 and TRF2 with mScarlet or mClover and examined their localization during mitotic arrest (Fig S2G). All proteins, including wild type (WT, WT-TRF1 and WT-TRF2) and the major phospho-null and phospho-mimetic mutants (TRF1-3A, TRF1-2D, TRF2-2A, and TRF2-2D), exhibited telomere-specific localization. These data are consistent with no effect of TRF1-S354/T358 and TRF2-S62/S65 phosphorylation on telomere

localization. We did not observe prominent localization to other genome locations but cannot rule out transient interactions. The data are described in lines 213-217 and 341-343.

2. The interconnection of P-TRF1 related functions with P-TRF2 related functions in MAD are not clear to me. What are overlapping functions? What functions specific for TRF1 or TRF2? Do P-TRF1 and P-TRF2 communicate with each other during MAD?

To address this insightful comment, we examined the phosphorylation state of TRF2-p65 in TRF1 depleted cells. TRF1 depletion resulted in reduced TRF2-pS65. This suggests that TRF1 phosphorylation promotes TRF2-pS65 phosphorylation and functional interdependence between the shelterin modifications. These data are consistent with a model where AURKB phosphorylates TRF1. Phosphorylated TRF1 then serves as binding site for the CPC component Survivin, which enables downstream TRF2 phosphorylation. The new data are presented in Fig. S4H, I and described in lines 345-347.

3. Given the focus on CPC in the title of the manuscript authors should better demonstrate the relevance of CPC components for MAD. CPC loss of function experiments (RNAi) with TIF analysis of metaphase arrested cells could be an option. In addition, RNAi mediated loss of individual CPC components may reduce TRF1 and TRF2 phosphorylation at the relevant sites. A possible phosphorylation (CPC) dependent interaction of BLM with TRF1 and/or TRF2 not clear to me. For example, it was not tested whether TRF2 wt and mutant versions can immunoprecipitate BLM and/or CPC components.

We thank the reviewer for their clear experimental suggestions.

RE: CPC. We used shRNAs targeting the CPC components INCENP or Survivin. Using live imaging we found that INCENP or Survivin depletion slightly reduced mitotic arrest in the shRNA treated cells (Fig. S3B). This was comparable to treatment with 40 nM of the AURKB inhibitor hesperadin (Fig. S3C-D). To maintain consistent mitotic arrest duration between experimental and control samples, we analysed MAD-TIF after 24 hours of colcemid treatment in INCENP or Survivin depleted cells and after 16-hours of colcemid for Control shRNA cells (Fig S3E). We observed a significant decrease in MAD-TIFs within the Survivin and INCENP depleted cultures.

These data compliment the in vitro and cell-based experiments demonstrating TRF1 and TRF2 are phosphorylated by the CPC component AURKB (Fig. 2D-E, 4B-C), and our observation that Survivin interacts with phosphorylated TRF1 (Fig 3C). Collectively the data are consistent with the CPC components INCENP, SURVIVIN, and AURKB all participating in MAD telomere deprotection. We discuss the new findings in lines 271-279.

RE: BLM interaction with TRF1 and TRF2. To examine these interactions in greater detail, we treated extracts from mitotically arrested cells with benzonase to degrade the genomic DNA prior to TRF1 or TRF2 pulldown. BLM recovery with TRF1 pulldown was benzonase insensitive, whereas BLM recovery with TRF2 pulldown was benzonase sensitive. This indicates TRF1 interacts directly with BLM independent of a DNA substrate, whereas a common DNA substrate (potentially the t-

loop junction) is required for TRF2 to interact with BLM. Neither TRF1 or TRF2 interactions with AURKB were benzonase sensitive, consistent with a direct interaction between the CPC and shelterin components. These new findings are presented in Figure S6E and discussed in lines 419-426. We did not explore the indirect BLM-TRF2 interaction in further detail.

RE: Potential phospho-regulation of TRF1-BLM interaction. At present we predict the TRF1-BLM interaction is phospho-independent. This is consistent with prior evidence of non-mitotic TRF1-BLM interactions (PMID: 25344324). In our experiments, phospho-mimetic and phospho-null TRF1 alleles pull down BLM to a similar degree from mitotically arrested extracts (Fig 3D). Likewise, both phosphorylated and unphosphorylated TRF1 peptides failed to significantly recover BLM from cell extracts (Fig S3A). However, phospho-null and -mimetic alleles only approximate the bona fide molecular states, and small TRF1 peptides may be incompatible with BLM binding. For these reasons we have elected to remain cautious in our interpretation and make no definitive conclusions in the manuscript. We discuss this in the discussion on lines 502-503

4. Data from TRF1 IP experiments (Fig. 3D) show that BLM interacts in an efficient manner also with mutant TRF1 versions, independently on the amount of AURKB in the 3immunoprecipitated. I am not clear about the role of the TRF1 (TRF2) phosphorylation for BLM recruitment. In addition, the role of other CPC components (Borealin, Survivin, TOP3A) for BLM interaction (or BRT function) is not clear.

We appreciate the reviewer's interest in this regulation between BLM and shelterin. Please see the response to point 2 and 3 above. Additionally, we note here:

- Our data is consistent with phosphorylation of TRF1 and TRF2 by the CPC component AURKB being essential for MAD-TIF. The CPC components INCENP and Survivin are also necessary for MAD-TIF, indicating the entire complex is required.
- TRF1 and BLM interact directly. TRF1 phosphorylation by AURKB, however, does not appear necessary for BLM recruitment.
- TRF2 and BLM interact through an intermediate DNA or protein substrate. We predict this is a t-loop junction, but did not explore this interaction further.

The BTR complex consists of BLM, TOP3A, and RMI1/2. We have not explored in detail but anticipate that the entire BTR complex is recruited to telomeres during MAD-telomere deprotection. This is consistent with the requirement of all complex members for MAD-TIF (Fig 6 and S6).

5. Very nice data was shown on telomere linearization in the part focusing on TRF2. However, the eventual impact of mutant TRF1 versions on telomere linearization remains unclear.

We appreciate the reviewer's comment and interest in the role of TRF1 in telomere macromolecular structure.

T-loop experiments are complicated by the limitation that only 15 – 30% of telomeres are recovered in a looped configuration. This is likely a physical limitation of the trioxsalen crosslinking required to maintain t-loop structure during sample preparation (discussed in depth in PMID: 30033372). In all experiments to date, telomere linearization corresponds with a pronounced induction of telomere deprotection (PMID: 24120135, 30033372, 31723267, 33239783, this manuscript). Unfortunately, subtle changes in protected/deprotected telomeres are insufficient to induce a significant change in the minor percentage of telomeres captured in looped configuration. This is statistical power limitation when only 15-30% of telomeres are captured in t-loops. For example, to induce sufficient MAD-telomere deprotection to confer a statistically significant change in telomere macromolecular structure, we relied on expressing TRF2 mutants (this manuscript) or a TRF2 shRNA (PMID: 30033372) to exacerbate the MAD-TIF phenotype to the greatest degree possible. Treating cells with Hesperadin suppressed MAD-TIF and prevented telomere linearization (PMID: 30033372), consistent with our conclusion that looped telomeres suppress DDR activation.

Unlike the TRF2 mutants that enhance MAD-TIF, TRF1 phospho-null mutants suppress MAD-TIF. Compared to WT-TRF1 where mitotic arrest confers ~ 10% deprotected telomeres, TRF1-3A reduces MAD-TIF to ~ 3% deprotected telomeres. Based on our collective expertise, this reduction in telomere deprotection will be insufficient to confer a significant change measured via t-loop assays. The most likely outcome of this experiment is a negative result due to the physical limitations and statistical power of the t-loop visualization.

Moreover, this experiment requires use of a murine system with longer telomeres. While the TRF2 regions of interest are conserved, the TRF1 regions of interest show greater variability between human and murine systems. Doing this experiment would first necessitate extensive preliminary characterization of murine TRF1 in the context of mitotic arrest. Finally, t-loop experiments are exceptionally laborious and time consuming. These difficulties are compounded in the context of mitotic arrest, which adds days of additional cell synchronization steps to the experimental procedure.

Given the technical challenges and limited feasibility of this approach, we concluded these experiments were beyond the revision scope and appreciate the reviewer's understanding.

6. General comment on TIF data on metaphase chromosomes: The representative images are very small and termini of individual chromosome cannot be identified. Thus, it is not really evident that DNA damage occurs exclusively at telomeres. Given that CPC activity is central for the manuscript, authors should make clear that damage is limited to telomeres or also occurring at other sites.

Thank you for the clear feedback. Following the reviewer's suggestion, we:

- Provided expanded images in Fig 1H showing clear γ -H2AX telomere co-localization.
- Quantified non-telomere γ -H2AX foci in the experimentation in Fig. 1G-I. This analysis confirmed that γ -H2AX foci induced by mitotic arrest are telomere specific and BLM-dependent.

These results are consistent with our prior reports that demonstrate mitotic arrest confers telomere-specific γ -H2AX foci (PMID: 22407014, 31530811). We now reiterate this point in the results. We acknowledge the importance of clear image representation and appreciate that with digital publication, readers can enlarge images as needed. Given space constraints, we refrained from adding additional enlarged views.

Other more specific questions related to individual figures of the manuscript:

7. Figure 1A: the APEX2-TRF1 is highly overexpressed in cells used for pull down experiments. Authors should comment eventual ectopic effects result in from TRF1 overexpression.

We agree with the reviewer that APEX2-TRF1 is highly overexpressed in the experiments in Figure 1. For this experiment, APEX2-TRF1 was overexpressed to facilitate unbiased interactomics. This was the preliminary step to identify potential interactors to be followed with mechanistic investigation for the remainder of the manuscript. We do not use the APEX2-TRF1 allele for any mechanistic experiments.

Importantly, for the mechanistic experiments in Fig 2 onward, we observed no significant difference in MAD-TIF between vector control and WT-TRF1 in the Control shRNA background (Fig 2B). This indicates that TRF1 overexpression does not impact MAD telomere deprotection.

8. Fig.1F: Authors chose BLM as main candidate for regulating MAD activity by binding to TRF1. Fig. 1F contains other candidates of interest – for example FBXO18 appears to be interesting as well. Other classic telomere regulators such as WRN or ATRX show up in the candidate list. Are there other arguments for focusing on BLM or excluding other candidates?

We thank the reviewer for highlighting the need to more clearly discuss candidates identified in Figure 1F. The focus of our study was identifying and characterizing factors involved in mitotic arrest dependent-telomere deprotection. In Figure 1F, the 0 hr time point corresponds to the G1/S border when cells were released from the double thymidine block. The 12, 14, and 16 hr time points correspond to 6, 8, and 10 hrs of mitotic arrest.

ATRX, FBXO18, DDX11 and others showed a strong TRF1 interaction at 0 hrs, consistent with a G1/early S telomere association. The association of these candidates with TRF1 diminished as a function of mitotic arrest (12 – 16 hr time points). This indicates the observed TRF1 interactions were interphase in nature.

Conversely, BLM had low TRF1 association at the G1/S border (0-hour) which increased as mitotic arrest progressed (12–16 hours). This is indicative of greater BLM-TRF1 interaction specifically during mitotic arrest. This piqued our interest in BLM potentially functioning at telomeres specifically during mitotic arrest.

We agree with the reviewer that WRN is an interesting candidate. Our team investigated the role of WRN in MAD-telomere deprotection in an independent study that is cited in this manuscript (PMID: 36635307, citation number 67 in this manuscript). That WRN, a RecQ family member, had

regulatory roles in MAD telomere deprotection also prompted our focus on the role of the RecQ BLM helicase in this study.

At the request of Reviewer 2, we have highlighted all helicases in Fig 1F that demonstrated enrichment during mitotic arrest. None of the other candidates are known telomere-specific regulators. These candidates will be the focus of future endeavors.

9. Figure 2 and 3: Experiment on the generated phosphor-specific TRF1 antibody and the phosphor-peptide pull down are very convincing. Nevertheless, it would be important to get an idea on the % of total TRF1 and also TRF2 protein subjected to phosphorylation by AURKB.

We appreciate the reviewer's interest in understanding the extent of TRF1 and 2 that is phosphorylated during mitotic arrest. This is a technically challenging experiment because the AURKB consensus sequence is highly basic and destroyed by commonly used proteases. This limits the utility of phospho-proteomics to identify and precisely quantify AURKB-dependent shelterin phosphorylation.

Despite multiple attempts, we could not devise a strategy to compare pan TRF1 or TRF2 blots to phospho-TRF1 or -TRF2 blots and compare signals to identify relative phosphorylation levels. We spent considerable effort to optimise phos-tag-based Western blotting to measure relative mobility of TRF1 or TRF2 in asynchronous and mitotically arrested cultures. While the data are generally consistent with TRF1 and 2 being phosphorylated in an AURKB-dependent manner in mitosis, the results are not clear enough to determine what percentage of TRF1 or 2 is phosphorylated. We have elected to exclude these data for that reason.

The key discovery in this work is the functional role of TRF1/2 phosphorylation in regulating MAD telomere deprotection. Knowing the exact proportion of phosphorylated TRF1 or TRF2 likely does not add substantial value at present.

10. Figure 2B and Supplementary figure 2D: What are the total levels of endogenous TRF1 and mutant TRF1 variants in the experimental cells? Suppl. Fig 2D shows only ectopic TRF1 levels. It would be important to know the proportion of wt TRF1 and mutant TRF1 in experimental cells (as shown for experiments with mutant TRF2 versions).

The reviewer identifies a needed clarification in our experimental set up.

For all mechanistic experiments using exogenous TRF1 we deplete the endogenous TRF1 with an shRNA while simultaneously expressing a shRNA-resistant exogenous TRF1 allele. The shRNA removes most of the endogenous protein. Therefore, the proportion of endogenous:exogenous TRF1 is heavily skewed towards the exogenous protein. This methodology is consistent with most experiments investigating shelterin function where the endogenous allele is deleted/depleted and complimented with over-expressed WT or mutant alleles (for example, see many papers from the de Lange or Boulton laboratories, and/or the following list of our prior publications PMID: 23850488, 30033372, 31723267, 33239783).

To confirm TRF1 depletion we blot against the endogenous proteins (Figure S2A). To detect exogenous FLAG-TRF1 and mutant derivatives we use an anti-FLAG antibody as there is less background compared to the commercial TRF1 antibody (Fig S2B, D, E).

While over-expressed, the expression of exogenous TRF1 remains relatively consistent between the different mutant alleles (Figure S2B, D, E). Additionally, while there is some variation between exogenous protein expression, this does not correlate with MAD-TIF outcomes. For example, WT-TRF1 is expressed to a lower level than the phospho-null alleles that suppress MAD TIF, and the phospho-mimetic alleles that promote MAD-TIF. Finally, as noted in point 7 above, there is no significant difference in MAD-TIF between vector control and WT-TRF1 in the Control shRNA background (Fig 2B). This indicates that TRF1 overexpression does not impact MAD telomere deprotection.

The experimental set up is clarified in lines 197 – 199 to aid reader comprehension.

11. Figure 2B: Results of metaphase TIF abundance are very nice. Given the importance of this data the impact of most important mutant TRF1 version should be validated in a different cell model (HT1080 for example).

Again, we thank the reviewer for a clear experimental suggestion that improved our manuscript. We conducted rescue experiments using TRF1 and the TRF1-T358A mutant in both HCT116 and HT1080 cell lines. Both cell models produced results consistent with those observed in IMR-90 E6E7 hTERT cells (Fig. S2D), supporting the reproducibility and robustness of our findings. The new finding is described in line 209-211.

12. Fig. 2D: Western blotting experiment was performed in TRF1 overexpression conditions. Can phosphorylation also be found on endogenous TRF1?

When we probe whole cell extracts from mitotically arrested cells using the TRF1-pT358 antibody there are multiple bands present (Fig S2E). This likely represents cross-reactivity with other phosphorylated AURKB targets present in mitosis. Unfortunately, the multiple bands present in whole cell extract complicate interpretation requiring the TRF1 immunoprecipitation step.

13. Additional question on data related to Figure 2:

a. Do the used TRF1 variants (Figure 2) locate to telomeres or are they located at other regions of chromosomes (IF, ChIP)

Please see our comment to Point 1 which addresses this concern.

b. TRF1 is a suppressor of replication stress and telomere fragility. Can authors show data on these features focusing on most relevant TRF1 mutants? Do these features potentially contribute to MAD? Is there DNA damage related to TRF1 mutants in interphase cells?

The reviewer is correct in the role of TRF1 in telomere replication stress and fragility. We appreciate their interest in a potential connection between fragile telomeres/replication stress and MAD-telomere deprotection. All our data are consistent with telomere fragility and MAD-telomere deprotection being independent phenotypes.

Specifically, TRF1 depletion induces telomere fragility but suppresses MAD-TIF formation. Conversely, MAD-TIF are induced during mitotic arrest in cells with wild-type shelterin where telomere fragility is suppressed. In this work, we did not observe an interphase DDR induced by TRF1 mutants nor telomere fragility. Finally, over many years studying this phenotype, we have consistently observed that MAD-TIF and fragile telomeres appear independently in mitotic spreads, with no detectable association between the two phenomena (PMID: 22407014, 23850488, 26108857, 31530811, and this work).

14. What is the final outcome of MAD impairment by most relevant TRF1 mutants on genome stability and cell survival? A possible readout is presented in Fig. 7 for the role of TRF2 on MAD.

We appreciate and understand the reviewers interest on TRF1 mutants on genome stability and cell survival. Our previous analyses indicate that increasing MAD-TIF results in elevated mitotic cell death and an expedited time to mitotic death (i.e. cells die more quickly during mitotic arrest; PMID: 26108857, 31530811). This underscores that telomere deprotection can promote mitotic death. However, MAD-telomere deprotection is not the primary driver of cell death during mitotic arrest. Cell lethality due to mitotic arrest primarily results from competition between Cyclin B1 degradation and activation of pro-apoptotic pathways (reviewed here PMID: 23890995 and here PMID: 29362479). This canonical mitotic death pathways is independent of MAD-telomere deprotection, and compensates when MAD-TIF are suppressed.

In agreement, suppressing MAD-TIF with TRF1 mutants failed to rescue mitotic death following treatment with replication stress or mitotic arrest inducing agents. This supports our prior observations that MAD-TIF are a minor mitotic lethality pathway, and that cells with suppressed MAD-TIF are still susceptible to mitotic lethality through the known primary apoptotic pathway. We elected to leave these data out of the manuscript as they do not substantially affect our conclusions.

On the contrary, increasing MAD-TIF can enhance and expedite mitotic death (PMID: 26108857). In response to reviewer 2, we have reproduced this result with TRF2 variants and a microtubule poison. These new data are included in the manuscript (Fig. 7) and discussed on lines 429-438.

We are currently investigating how MAD-TIF promote mitotic death. However, this is a substantial project outside the scope of this manuscript.

15. Fig. 3D: TRF2 appears to bind with reduced efficacy to mutant TRF1 versions TRF1deltaHD and TRF1-2D. TRF2 binding to TRF1-2A appears to be normal. It would be interesting to validate this finding by ChIP on telomere repeat containing chromatin as this may have an impact on the formation of linear telomeres.

Thank you for this keen observation. While we agree this could provide interesting insights into shelterin dynamics, we conclude that relative levels of TRF1-TRF2 interaction do not impact MAD-telomere deprotection.

In support of this interpretation, there is no consistent correlation between the MAD-TIF phenotype and TRF1-TRF2 interactions. For instance, both WT TRF1 (which rescues MAD-TIF) and TRF1-2A (which fails to rescue MAD-TIF) exhibit enhanced TRF1-TRF2 interactions, whereas TRF1 Δ HD (which fails to rescue MAD-TIF) and TRF1-2D (which rescues MAD-TIF) show reduced TRF1-TRF2 interactions. This variability suggests that TRF1-TRF2 binding levels are not directly tied to the MAD-TIF response. Because of this observation, and due to the phospho-specific antibodies being incompatible with ChIP assays, we did not pursue the suggested experiment.

16. Figure 4F, Supplementary Figure 4E: Do the mutant TRF2 versions locate to telomeres or other locations along chromosomes? Does the expression of TRF2 variant sequences have an impact of TRF1 localization interphase cells or metaphase cells?

Please see our comment to Point 1 which addresses this concern.

17. Supplementary figure 4A. I do not understand why the pTRF2(S65) band in lane 10+11 runs higher than in other lanes (top blot).

We appreciate the reviewers comment as there are closely set bands that cause some confusion. We show here that the pTRF2-S65 band (red line) is present in the mitotically arrested samples in Lane 4, 5, 6, 10, and 11.

A second lower molecular weight band is present in lanes 1-4, 8 and 9 (blue line). Given that these faster-migrating bands are still detected in the shTRF2 condition (lane 2), we consider them to be non-specific bands.

18. Western blotting experiment was performed in TRF2 overexpression conditions. Can phosphorylation also be found on endogenous TRF2?

In brief, no, we were unable to detect the phospho-specific band on endogenous TRF2. However, endogenous TRF2 is not highly expressed, and phospho-specific bands are often difficult to detect relative to the endogenous blots using pan antibodies. It is not surprising this was difficult and is likely a result of relative expression and phosphorylation levels.

We note the phospho-specific band is present on myc-TRF2 without immunoprecipitation (see lane 4 in the blots presented in point 17). Enriching with IP facilitated quantitative assessment for Fig 4B.

19. Supplementary Figure 4C: TRF2 levels shown by western blotting appear to be heterogeneous in the experimental cell lines. Can this impact on DNA damage data shown later in the manuscript?

The reviewer is correct regarding heterogeneous TRF2 expression. However, we are not concerned.

Specifically, all exogenous alleles regardless of expression level suppressed interphase TIF (Figure S4F). This is consistent with functional interphase telomere protection, indicating all observed results are specifically related to MAD-TIF. Further, there is no direct correlation between TRF2 expression levels and MAD-TIF abundance. High TRF2-expressing constructs include TRF2-62A that exhibits low MAD-TIF and TRF2-2D that shows high MAD-TIF; and low TRF2-expressing constructs include TRF2-2A that exhibits low MAD-TIF and TRF2-S65D that exhibits high MAD-TIF. Finally, our previous published data indicate that TRF2 levels can be reduced by up to 90% without inducing TIF (PMID: 23850488) suggesting that expression level alone is not a determining factor for the observed phenotype. All data align on the regulation of MAD-TIF being a function of TRF2 phosphorylation status on S62 and S65, not relative expression levels.

20. Fig. 4F; Supplementary figure 4G: Given the importance of the demonstration of TRF2 phosphorylation on mitotic TIF formation, it is suggested to validate most relevant TRF2 variants in an independent cell model (HT1080).

We agree with the reviewer regarding the critical need to validate results across cell lines. This was required for TRF1 and addressed above (Point 11). TRF2 phosphorylation, however, is tested in this paper under three distinct conditions: IMR-90 with colcemid treatment (Fig 4), murine embryonic fibroblasts with nocodazole treatment (Fig 5), and HT1080 6TG cells with aphidicolin-induced mitotic arrest (Fig S7). Across all models, we observed that the two phosphorylation sites in TRF2 play a crucial role in suppressing MAD-TIF, strengthening our conclusion.

21. Supplementary figure 4G: The effect of ectopically expressed, wt TRF2 is missing in the quantification of TIF data – that makes it difficult to get an idea on the effect of the tested TRF2 mutant proteins.

We thank the reviewer for noticing these excluded data. We have now added the TRF2-WT data to Supplementary Figure 4G, using the same data presented in Figure 4F. We have also clarified this in the figure legend. We note including the TRF2-WT data did not alter panel conclusions.

22. In addition, TIF data of all samples appears to be highly variable at the 24hrs time point. Numbers of metaphase should be increased to provide more convincing data.

RE: Variability. We previously demonstrated the amount of MAD-TIF is directly proportional to mitotic duration (PMID: 22407014, 30033372). In these prior experiments, we showed tight concordance in MAD-TIF when we held cells to specific mitotic arrest duration (e.g. 6, 10, or 14 hrs of mitotic arrest; PMID: 22407014 Fig 1E; PMID: 30033372 Fig S6D). We did this by synchronizing cells in G1/S with a double thymidine block, releasing into S-phase in the presence of a microtubule poison, allowing the cells to enter mitosis, and then collecting samples for MAD-TIF experiments within a specific temporal window of mitotic arrest. This is a time consuming 5-day experiment, with time points difficult to coordinate during a normal working day.

To simplify experimentation, here we treated asynchronous cultures with colcemid for 2 or 24 hours. Two hours of colcemid is insufficient to induce MAD-TIF, while 24 hours induces significant MAD-TIF (Fig. 1I). The caveat of this approach is that a range of mitotic arrest durations are examined based on when the individual cell in question entered mitosis. For 24-hour colcemid samples this can range from 24 hours (long mitotic arrest = high MAD-TIF) to < 2 hrs (short mitotic arrest = low MAD-TIF). This is why there is a high variability in the total MAD-TIF number. We note that in practice most cells perish by 14 hours of mitotic arrest. By simplifying our experimental treatment, we reduced the experiment duration to two days, lowered research costs, and increased our rate of data collection. We have edited the text to clarify that MAD-TIF is directly proportional to mitotic duration on line 76-77 and 181-182.

Re: Metaphase Numbers. In its current form, the paper includes 98 experimental conditions examined by metaphase-TIF analysis, all completed in triplicate: $98 * 3 = 294$ individual meta-TIF replicates. We quantify 20+ metaphases per replicate; $20 \text{ metaphases} * 294 \text{ replicates} = 5,880$ mitotic cells quantified. Each cell has 46 chromosomes = 92 chromosome ends. Following genome duplication in S-phase, a mitotic cells contains 184 telomeres. For this project we scored at least $5,880 \text{ metaphases} * 184 \text{ telomeres} =$ a conservative estimate of 1,081,920 telomeres scored. We hope this assuages the reviewer's concern.

Reviewer #2

Mitotic arrest that is accompanied by a partial telomere deprotection has been reported since a while. Intriguingly, this partial deprotection involved resolution of the t-loops in a telomere length independent manner and did trigger an ATM-dependent DNA damage response. Yet, it does not yield telomere rapid deletion events or end-to-end fusions, which would be hallmarks of telomere deprotection. This latter classically was ascribed to the telomere binding protein TRF2 only. However, it remained unclear how the mitotic arrest dependent (MAD) phenomenon is triggered inside cells and what factors are associated with it.

Using APEX methodology combined with proteomics, here it is shown that this MAD partial deprotection involves two major complexes, first the chromosome passenger complex (CPC, with the AURKB kinase as executing enzyme). The AURKB was already known to be somehow involved and the authors then go on to assess the functional importance and the targets of AURKB, which remarkably yielded TRF1 and TRF2. Specific phosphorylation sites were mapped *in vitro* / *in vivo*, and P-specific antibodies allowed further refinement of the mechanistic impact of those phosphorylations. This in turn led to the discovery of, secondly, the BTR complex as being associated with MAD telomeres and most likely the agent that mediates the partial deprotection.

These results advance our understanding of telomere biology in several very important ways: the results allow the description of a pathway of TRF1-P that attracts the CPC and allows phosphorylation of TRF2 and a specific type of partial telomere deprotection (Fig. 7g). Secondly, the data postulate that both, TRF1 and TRF2 are involved in this deprotection pathway, which at the same time is a partial protection pathway (re no rapid deletions and end-fusions). The TRF1 protein in this context is required to bring the CPC with the AURKB to the telomeres and without that, TRF2 is not phosphorylated and does not elicit a MAD-TIFs. Therefore, the interplay between the shelterin proteins TRF1 and TRF2 is more intricate than anticipated and previously stipulated.

The experiments are extremely well designed and executed at an amazing level of technical quality and precision. This reviewer also appreciated the efforts to describe the sometimes quite involved setups in a clear and transparent way. To me, this manuscript clearly is breaking new ground on several fronts and deserves a very high visibility.

There are only a few minor comments / suggestions for corrections.

We thank the reviewer for their thorough reading of our manuscript, and for their kind comments and support. We have addressed each of their comments below.

1) Fig 1D and S1G: line 150 ff in the text says that "...the CPC components AURKB, INCENP, and CDCA8 (Borealin), were strongly enriched within the Flag-APEX2-TRF1 samples ". Upon inspection of the data, I could not validate the claim for AURKB: it seems to be in the non-specific domain in the volcano plot and barely distances itself from baseline in Fig. S1G. AURKB has been identified previously for this pathway (Ref 25) and it makes sense to follow that up. But the description here does not match the data.

Thank you for your careful review and for pointing out this discrepancy. We have revised the text to more accurately reflect the data in Figure 1D and Figure S1G, ensuring that it aligns with the

observed enrichment levels in the volcano plot (lines 154-159). We appreciate your attention to detail and have made the necessary adjustments to the manuscript accordingly.

2) While I agree that the mitotic death phenotype induced by MAD-TIF is a significant read-out for the consequences of this effect, I am not enthusiastic about the way the experiments were done in Fig. 7. TIF induction is by lethal dose of aphidicolin, which is a DNA polymerase inhibitor and well known for its effects on S-phase cells. Therefore, as the authors acknowledge in the text, the vast majority of TIFs scored in this experiment probably has nothing to do with MAD-cell death but is associated with S-phase catastrophe. Therefore, I am unsure whether the cell-death suppression phenotype in Fig. 7E is related to MAD-cell death or not. I would suggest performing this experiment in a genuine MAD background (microtubule inhibitor for example) such that we can be more confident of the MAD associated cell death.

This is an astute observation, and we agree with the reviewer that the additional experiments using a mitotic inhibitor are necessary.

To address this concern, we performed live imaging of shTRF2 expressing IMR-90 E6E7 hTERT complimented with vector, TRF2-WT, TRF2-S65A, or TRF2-S65D. The cells were treated with colcemid to induce mitotic arrest. Consistent with our previous findings (PMID: 26108857), shTRF2 induced a significant increase early mitotic death between 2–6 hours of mitotic arrest. Importantly, we observed that TRF2-WT and TRF2-S65A, which inhibit MAD-TIF, suppressed this mitotic cell death. Conversely, TRF2-S65D which promotes MAD-TIF, did not. This is consistent with a MAD-TIF promoting mitotic death.

These new results reinforce our hypothesis and providing a direct link between MAD-TIF and mitotic death. We have now included this data as a new Figure 7 in the main manuscript (lines 429-438), and we have moved the original aphidicolin data to Supplementary Figure S7.

Minor issues:

3) Please name TRF1 and TRF2 consistently: in volcano plot and in S1E they are labeled TERF1 and TERF2.

We have edited as requested.

4) Abstract lines 20-32 are a very difficult read. Please break up sentence for clarity.

We agree and thank the reviewer for suggesting. This sentence was broken up as suggested.

5) Perhaps mark the 11 helicases in Fig. 1F and mentioned in the text (line 166).

We agree with this suggested and have edited text and figure labels accordingly.

Other questions that I feel are pertinent but that do not need to be addressed with experiments:

- Why/how is TRF1-p by AURKB specific to M-phase?

We think this specificity is due to the strict regulation of AURKB activity throughout the cell cycle, with peak activity occurring during M-phase. The mechanism by which telomeric TRF1 is phosphorylated by AURKB, which is primarily localized at centromeres as part of the CPC complex, is indeed intriguing. One possibility is that a small fraction of TRF1 undergoes phosphorylation through initial, stochastic interactions with CPC, facilitated by random diffusion. Phosphorylation of the myb domain in TRF1 could then disrupt its telomeric localization (FL Chan et al. 2017), promoting further phosphorylation in a feed-forward manner. We discuss these points in the Discussion (line 486-493).

- Could you do pull-back experiments to evaluate when the MAD arrest and cell death phenotype become irreversible? I.e. which is the execution step for the cell death.

This is an insightful suggestion. Identifying the point at which mitotic arrest and cell death become irreversible is indeed critical, and it is an area we are actively investigating.

Our current understanding suggests that the primary mechanism of cell death involves cohesin fatigue leading to apoptotic death, with MAD telomere deprotection acting as a backup mechanism (PMID: 31530811). To pinpoint the execution step for cell death specifically linked to telomere MAD-TIF, we need to develop a live-cell reporter for more targeted detection of telomeric MAD-TIFs. We are currently working to establish such methods, though this is beyond the scope of the present study.